# ADAPTIVE CAUSAL EXPERIMENTAL DESIGN: AMORTIZING SEQUENTIAL BAYESIAN EXPERIMENTAL DESIGN FOR CAUSAL MODELS

## ABSTRACT

Interventions are essential for causal discovery and causal reasoning. Acquiring interventional data, however, is often costly, especially in real-world systems. A careful experimental design can therefore bring substantial savings. In a sequential experimental design setting, most existing approaches seek the best interventions in a greedy (myopic) manner that does not account for the synergy from yet-to-come future experiments. We propose Adaptive Causal Experimental Design (ACED), a novel sequential design framework for learning a design policy capable of generating non-myopic interventions that incorporate the effect on future experiments. In particular, ACED maximizes the Expected Information Gain (EIG) on flexible choices of causal quantities of interest (e.g., causal graph structure, and specific causal effects) directly, bypassing the need for computing intermediate posteriors in the experimental sequence. Leveraging a variational lower bound estimator for the EIG, ACED trains an amortized policy network that can be executed rapidly during deployment. We present numerical results demonstrating ACED's effectiveness on synthetic datasets with both linear and nonlinear structural causal models, as well as on in-silico single-cell gene expression datasets.

## 1 INTRODUCTION

Identifying and modeling causal relationships are of fundamental importance across various scientific disciplines, including biology (Tejada-Lapuerta et al., 2023), medicine (Sanchez et al., 2022), economics (Varian, 2016), and social science (Sobel, 2000; Imbens, 2024). A key step of this process involves learning the underlying causal model, often represented as a Structural Causal Model (SCM). A SCM consists of a directed acyclic graph (DAG) that captures the causal connections among variables, and a set of conditional distributions that quantifies their probabilistic dependencies. When the causal graph is unknown, causal discovery aims to learn the causal structure from data. With only observational data and no assumption about the data-generating process, causal discovery methods can only recover DAGs up to their Markov equivalence class (MEC) (Verma & Pearl, 2022). This limitation has motivated the use of interventional data that can disambiguate causal relationships and improve the causal graph's identifiability (Pearl, 2009). However, interventional experiments are often time-consuming, expensive, and sometimes ethically problematic. It is therefore desirable to systematically design these experiments in order to maximize their value. Bayesian optimal experimental design (BOED) has emerged as a powerful framework to achieve this goal (Lindley, 1956b; Chaloner & Verdinelli, 1995; Rainforth et al., 2024; Huan et al., 2024).

Most existing BOED methods for causal models (Cho et al., 2016; Ness et al., 2018; Agrawal et al., 2019; Tigas et al., 2022; 2023) essentially perform active learning on SCMs (Figure 1a). This active learning procedure involves iteratively designing interventions with the highest expected information gain (EIG), acquiring interventional data under the selected design, and performing Bayesian inference over the SCM. While effective, these methods have three significant limitations. (1) *Myopic intervention design*: they optimize the next intervention without accounting for future experiments yet to come, leading to suboptimal intervention decisions over the entire design horizon. (2) *High computational costs at deployment*: at each experiment, they require online computations of the posterior update and intervention optimization. (3) *Inefficiency and suboptimality in learning a full causal model for targeted causal queries*: when only specific causal effects are of interest, targeting

(a) Existing active causal BOED based on a myopic formulation.

(b) ACED trains a policy $\pi$ offline, which can be deployed for fast online usage to produce adaptive, non-myopic interventions.

Figure 1: Comparison of existing active causal BOED and proposed ACED. Here $\mathcal{D}$ denotes the observation data before performing interventions, at each experimental stage $t$, a set of intervention data $\mathcal{D}_{int}$ could be collected for belief update.

experiments to learn the entire causal graph becomes inefficient and leads to suboptimal intervention decisions (Smith et al., 2023). Indeed, the ultimate goal often is to enable causal reasoning on specific quantities of interest (QoI)—answering causal queries, estimating treatment effects, and addressing counterfactual questions—instead of the causal graph itself. For example in biology, breast cancer experiments are usually motivated by causal relationships between specific oncogenes and tumor suppressor genes, rather than the entire gene regulatory network.

To overcome these limitations, we propose Adaptive Causal Experimental Design (ACED), a novel approach that employs a transformer-based policy network that takes designs and data from completed experiments as inputs, and outputs the intervention design for the next experiment (Figure 1b). ACED provides three key contributions.

- **Non-myopic intervention designs.** The policy network is trained by maximizing the total EIG of the entire sequence of experiments rather than only on the upcoming experiment. Notably, under ACED, the total EIG can be estimated without evaluating intermediate posteriors, thus sidestepping the need for repeated Bayesian inference.
- **Real-time adaptive decision-making.** By training a single amortized policy network offline, ACED enables rapid, real-time decision-making, significantly reducing online computational needs at deployment.
- **Flexible causal queries.** ACED can accommodate a wide range of causal-related design goals, including causal discovery and causal reasoning, allowing experiments to focus on improving specific causal QoIs.

## 2 RELATED WORK

BOED (Lindley, 1956a; Foster et al., 2019; Rainforth et al., 2024; Huan et al., 2024) seeks to identify experiments that generate the most informative data, often measured by the EIG of model parameters (equivalently, mutual information between model parameters and data). The EIG is generally intractable and needs be estimated numerically, for example via nested Monte Carlo (Ryan, 2003; Huan & Marzouk, 2013) and variational lower bounds (Foster et al., 2019; Kleinegesse & Gutmann, 2020). In the context of causal discovery, BOED has been specifically applied to identify intervention targets and values, which serve as experimental conditions. This has been

explored in several works (Cho et al., 2016; Ness et al., 2018; Agrawal et al., 2019; Tigas et al., 2022), demonstrating its utility in selecting interventions that improve the identifiability of causal structures. More recently, Reinforcement Learing (RL) based methods have been used to learn non-myopic policies that maximize the EIG over an entire sequence of experiments (Foster et al., 2021; Ivanova et al., 2021; Blau et al., 2022; Shen & Huan, 2023), and have also been applied for sequential causal discovery (Annadani et al., 2024a; Gao et al., 2024a). While most BOED approaches focus on maximizing the EIG of the model parameters, it has been shown that when the goal is to answer specific causal queries—such as predicting causal effects or reasoning about counterfactuals—optimizing the EIG for the quantities of interest (QoI) directly can lead to better outcomes (Bernardo, 1979; Attia et al., 2018; Wu et al., 2021; Smith et al., 2023). Toth et al. (2022) demonstrated the effectiveness of this approach for various causal tasks, including causal discovery, causal inference, and causal reasoning, in a myopic design setting. We include additional details on the related work in Appendix 9.

## 3 BACKGROUND

### 3.1 STRUCTURAL CAUSAL MODEL

Let $\boldsymbol{V} = \{1, \ldots, d\}$ be the vertex set of a graph $G = \{\boldsymbol{V}, E\}$ and $\boldsymbol{X} = \{X_1, \ldots, X_d\} \subseteq \mathcal{X}$ be the associated random variables. A SCM includes $G$ and associated parameters $\boldsymbol{\theta} = \{\boldsymbol{\theta}_1, ..., \boldsymbol{\theta}_d\}$ is defined by:

$$X_i = f_i(\boldsymbol{X}_{par(i)}, \boldsymbol{\theta}_i; \epsilon_i), \quad \forall i \in \boldsymbol{V}, \tag{1}$$

where $f_i$ is the causal mechanisms from the parent nodes of $X_i$, denoted by $\boldsymbol{X}_{par(i)}$, to $X_i$ with parameters $\boldsymbol{\theta}_i$ governing the relationship between $X_{\text{par}(i)}$ and $X_i$, and $\epsilon_i$ are exogenous noise variables.

A perfect intervention on $X_i$ is denoted by $\mathrm{do}(X_i = s_i)$ (Pearl, 2009), which sets the node $X_i = s_i$. We encode the intervention as design variable $\boldsymbol{\xi} = \{\mathcal{I}, s_{\mathcal{I}}\}$ where $\mathcal{I}$ is the intervention node index. Assuming causal sufficiency and independent noise (Spirtes et al., 2000), the likelihood of data $X$ given $\boldsymbol{\theta}$ follows the Markov factorization:

$$p(\boldsymbol{X} \mid G, \boldsymbol{\theta}, \boldsymbol{\xi}) = \prod_{j \in \boldsymbol{V} \setminus \mathcal{I}} p(X_j \mid \boldsymbol{X}_{par(j)}, \boldsymbol{\theta}_j, \mathrm{do}(X_{\mathcal{I}} = s_{\mathcal{I}})). \tag{2}$$

### 3.2 BAYESIAN OPTIMAL EXPERIMENTAL DESIGN FOR TARGETED CAUSAL QUERIES

While causal discovery and reasoning are typically handled as separate, consecutive processes, we present a unified framework to design the most informative sequence of interventions targeting causal QoIs under a fixed budget of $T$ experiments in an adaptive, non-myopic manner. We denote the general QoIs based on the causal model as $\boldsymbol{Z} = H(G, \boldsymbol{\theta}; \epsilon_{\boldsymbol{Z}})$. Here, $\boldsymbol{Z}$ is a specific predictive quantity based on $G$ and $\boldsymbol{\theta}$, and we assume $\boldsymbol{Z}$ is conditionally independent to $\boldsymbol{X}$ given $G$ and $\boldsymbol{\theta}$. For example, selecting $\boldsymbol{Z} = G$ corresponds to *causal discovery*, and $\boldsymbol{Z} = X_i^{\mathrm{do}(X_j = \psi_j)}$ represents *causal reasoning* which refers to the effect on $X_i$ by setting $X_j = \psi_j$, potentially with random $\psi_j \sim p(\psi_j)$. Let $\mathcal{D}$ denote the pre-existing data (if any) before performing any experiment, and let $\boldsymbol{h}_t = \{\boldsymbol{\xi}_{1:t}, \boldsymbol{x}_{1:t}\}$ denote the completed designs and interventions so far in the current sequence of experiments. Then, the belief on $G$ and $\boldsymbol{\theta}$ are updated following Bayes' rule:

$$p(G|\mathcal{D}, \boldsymbol{h}_t) = \frac{p(G|\mathcal{D}, \boldsymbol{h}_{t-1})p(\boldsymbol{x}_t|\mathcal{D}, G, \boldsymbol{h}_{t-1}, \boldsymbol{\xi}_t)}{p(\boldsymbol{x}_t|\mathcal{D}, \boldsymbol{h}_{t-1})}, \tag{3}$$

$$p(\boldsymbol{\theta}|\mathcal{D}, G, \boldsymbol{h}_t) = \frac{p(\boldsymbol{\theta}|\mathcal{D}, G, \boldsymbol{h}_{t-1})p(\boldsymbol{x}_t|\mathcal{D}, G, \boldsymbol{\theta}, \boldsymbol{h}_{t-1}, \xi_t)}{p(\boldsymbol{x}_t|\mathcal{D}, G, \boldsymbol{h}_{t-1})}, \tag{4}$$

where $p(\boldsymbol{x}_t|\mathcal{D}, G, \boldsymbol{h}_{t-1}, \boldsymbol{\xi}_t) = \int_{\boldsymbol{\theta}} p(\boldsymbol{x}_t|\mathcal{D}, G, \boldsymbol{\theta}, \boldsymbol{h}_{t-1}, \boldsymbol{\xi}_t)p(\boldsymbol{\theta})\, \mathrm{d}\boldsymbol{\theta}$ involves marginalization over $\boldsymbol{\theta}$. The corresponding prior-predictive and posterior-predictive densities for $\boldsymbol{z}$ are respectively:

$$p(\boldsymbol{z}|\mathcal{D}, \boldsymbol{h}_{t-1}) = \sum_G \int_{\boldsymbol{\theta}} p(\boldsymbol{z}|G, \boldsymbol{\theta}, \boldsymbol{h}_{t-1})p(G|\mathcal{D}, \boldsymbol{h}_{t-1})p(\boldsymbol{\theta}|\mathcal{D}, G, \boldsymbol{h}_{t-1})\, \mathrm{d}\boldsymbol{\theta}, \tag{5}$$

$$p(\boldsymbol{z}|\mathcal{D}, \boldsymbol{h}_{t-1}, \boldsymbol{x}_t, \boldsymbol{\xi}_t) = \sum_G \int_{\boldsymbol{\theta}} p(\boldsymbol{z}|G, \boldsymbol{\theta}, \boldsymbol{h}_t)p(G|\mathcal{D}, \boldsymbol{h}_t)p(\boldsymbol{\theta}|\mathcal{D}, G, \boldsymbol{h}_t)\, \mathrm{d}\boldsymbol{\theta}. \tag{6}$$

Therefore, the belief about QoI $\boldsymbol{Z}$ is updated indirectly through belief updates about $G$ and $\boldsymbol{\theta}$ based on new data. When $\boldsymbol{Z}$ is a one-to-one mapping of $G$ and $\boldsymbol{\theta}$, the EIG on $\boldsymbol{Z}$ equals the EIG on $G$ and $\boldsymbol{\theta}$ Bernardo (1979). In more general non-invertible cases, directly maximizing EIG on $\boldsymbol{Z}$ could be more efficient, as it focuses on resolving uncertainty relevant to $\boldsymbol{Z}$ only while avoiding unnecessary complexity from $G$ and $\boldsymbol{\theta}$. To quantify the informativeness of provided by an experiment, we adopt the EIG on $\boldsymbol{Z}$:

$$I_t(\boldsymbol{\xi}_t) = \mathbb{E}_{p(\boldsymbol{x}_t|\mathcal{D}, \boldsymbol{h}_{t-1}, \boldsymbol{\xi}_t)p(\boldsymbol{z}|\mathcal{D}, \boldsymbol{h}_{t-1}, \boldsymbol{x}_t, \boldsymbol{\xi}_t)}[\log \frac{p(\boldsymbol{z}|\mathcal{D}, \boldsymbol{h}_{t-1}, \boldsymbol{x}_t, \boldsymbol{\xi}_t)}{p(\boldsymbol{z}|\mathcal{D}, \boldsymbol{h}_{t-1})}]. \tag{7}$$

## 4 Adaptive Causal Experimental Design

We introduce ACED, which trains a policy $\pi$ to map from $\boldsymbol{h}_{t-1}$ to $\boldsymbol{\xi}_t$ in order to maximize the total EIG over the entire sequence of experiments.

**Proposition 1.** *The total EIG of a policy $\pi$ on QoI over a sequence of T experiments is*

$$\mathcal{I}_T(\pi) = \mathbb{E}_{p(G, \boldsymbol{\theta}|\mathcal{D})p(\boldsymbol{h}_T|\mathcal{D}, G, \boldsymbol{\theta}, \pi)} \left[ \sum_{t=1}^{T} I_t(\boldsymbol{\xi}_t) \right]$$

$$= \mathbb{E}_{p(G, \boldsymbol{\theta}|\mathcal{D})p(\boldsymbol{h}_T|\mathcal{D}, G, \boldsymbol{\theta}, \pi)p(\boldsymbol{z}|\mathcal{D}, G, \boldsymbol{\theta})} \left[ \log \frac{p(\boldsymbol{z}|\mathcal{D}, \boldsymbol{h}_T)}{p(\boldsymbol{z}|\mathcal{D})} \right]. \tag{8}$$

A proof is provided in Appendix 7.1. As shown in Shen et al. (2023), for the optimal policy, $\mathcal{I}_T(\pi^*) \geq \mathcal{I}_T(\pi_{\text{greedy}})$, where $\mathcal{I}_T(\pi_{\text{greedy}})$ refers to the greedy design policy. Notably, intermediate posteriors do not appear in this objective. Since the total EIG generally cannot be evaluated in close form, we approach it via a lower bound estimator. Denoting $\mathcal{M} = \{G, \boldsymbol{\theta}\}$ for simplicity, the lower bound is formed by replacing the true posterior $p(\boldsymbol{z}|\mathcal{D}, \pi, \boldsymbol{h}_T)$ with $q_{\boldsymbol{\lambda}}(\boldsymbol{z}|\mathcal{D}, \pi, f_{\boldsymbol{\phi}}(\boldsymbol{h}_T))$ parameterized by $\boldsymbol{\lambda}$, which is known as the Barber-Agakov lower bound Barber & Agakov (2004):

$$\mathcal{I}_{T; L}(\pi) = \mathbb{E}_{p(\mathcal{M}|\mathcal{D})p(\boldsymbol{h}_T|\mathcal{D}, \mathcal{M}, \pi)p(\boldsymbol{z}|\mathcal{D}, \mathcal{M})} \left[ \log \frac{q_{\boldsymbol{\lambda}}(\boldsymbol{z}|\mathcal{D}, f_{\boldsymbol{\phi}}(\boldsymbol{h}_T))}{p(\boldsymbol{z}|\mathcal{D})} \right]. \tag{9}$$

**Theorem 1.** *(Variational Lower Bound) For any policy $\pi$, variational parameter $\boldsymbol{\lambda}$ and data embedding parameter $\boldsymbol{\phi}$, $\mathcal{I}_T(\pi) \geq \mathcal{I}_{T; L}(\pi)$. The bound is tight if and only if $p(\boldsymbol{z}|\mathcal{D}, \pi, \boldsymbol{h}_T) = q_{\boldsymbol{\lambda}}(\boldsymbol{z}|\mathcal{D}, \pi, f_{\boldsymbol{\phi}}(\boldsymbol{h}_T))$ for all $\{\boldsymbol{z}, \boldsymbol{h}_T\}$ and $\pi$.*

A proof is provided in Appendix 7.1. Here $f_{\boldsymbol{\phi}}(\boldsymbol{h}_T)$ is an embedding of $\boldsymbol{h}_T$. Thus the lower bound of EIG on a policy can be tightened via maximizing $I_{T; L}(\pi)$ with respect to $\boldsymbol{\lambda}$ and $\boldsymbol{\phi}$. As the denominator is constant with respect to $\boldsymbol{\lambda}$ and $\pi$, it can be omitted, leading to a simplified objective

$$\pi^*, \lambda^*, \phi^* = \arg\max_{\pi, \lambda, \phi} \mathcal{R}_T(\pi), \tag{10}$$

where $\mathcal{R}_T(\pi) = \mathbb{E}_{p(\mathcal{M})p(\boldsymbol{h}_T|\mathcal{M}, \pi)}[\log q_{\boldsymbol{\lambda}}(\boldsymbol{z}|\mathcal{D}, f_{\boldsymbol{\phi}}(\boldsymbol{h}_T))]$ is an EIG lower bound shifted by the constant denominator.

### 4.1 Variational Posterior for Targeted Causal Queries

To select an appropriate variational family for $q_{\boldsymbol{\lambda}}(\boldsymbol{z}|\mathcal{D}, f_{\boldsymbol{\phi}}(\boldsymbol{h}_T))$, we base our choice on the support of specific $\boldsymbol{Z}$, considering whether it is continuous, discrete, or bounded. For example, when dealing with causal discovery $\boldsymbol{Z} = G$, we follow Lorch et al. (2022) and model the existence of an edge $G_{i,j}$ using independent Bernoulli distribution, that is

$$q_{\boldsymbol{\lambda}}(G|\mathcal{D}, f_{\boldsymbol{\phi}}(\boldsymbol{h}_T)) = \prod_{i,j} q_{\boldsymbol{\lambda}}(G_{i,j}|\mathcal{D}, f_{\boldsymbol{\phi}}(\boldsymbol{h}_T)) \quad \text{with} \quad G_{i,j} \sim \text{Bernoulli}(\boldsymbol{\lambda}_{i,j}). \tag{11}$$

For causal reasoning $\boldsymbol{Z} = X_i^{\text{do}(X_j = \psi_j)}$, since the posterior belief on $\boldsymbol{Z}$ captures belief on multiple graphs, and thus could be potentially multi-modal. Therefore, we adopt Normalizing Flows (NFs), which is a type of generative model that uses a series of invertible mappings to transform from a simple distribution (i.e., standard normal) to a general target distribution. Specifically, we use the real NVP (Dinh et al., 2016) architecture for representing $q_{\boldsymbol{\lambda}}(\boldsymbol{z}|\mathcal{D}, f_{\boldsymbol{\phi}}(\boldsymbol{h}_T))$, with details in Appendix 7.2.

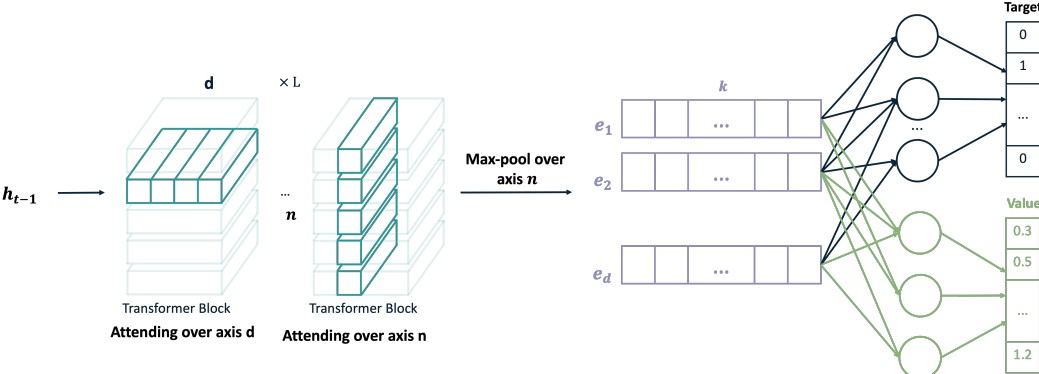

Figure 2: Policy network architecture. Our model maps the input to a three-dimensional tensor of shape $n \times d \times 2$ and remains permutation in- and equivariant over axes $n$ and $d$, respectively. Each of the L layers first self-attends over axis $d$ and then over $n$, sharing parameters across the other axis.

## 4.2 POLICY NETWORK AND EMBEDDING ARCHITECTURE

The policy network $\pi$ is summarized in Figure 2 and is designed to satisfy key symmetries inherent to BOED and causal structure learning: permutation invariance across $n$ history samples and permutation equivariance across $d$ variables. The core of $\pi$ consists of $L$ identical layers, each comprising four residual sublayers: two multi-head self-attention sublayers alternating with two position-wise feed-forward network sublayers, resembling the Transformer encoder architecture (Vaswani, 2017). To enable information flow across all $n \times d$ tokens, the model alternates attention over observation and variable dimensions (Kossen et al., 2021). The first self-attention sublayer attends over the variable axis $d$, while the second attends over the sample axis $n$, ensuring representation equivariance (Lee et al., 2019). After building the representation tensor, max-pooling over the observation axis $n$ yields a representation $(\boldsymbol{e}^1, \ldots, \boldsymbol{e}^d)$, where each $\boldsymbol{e}^i \in \mathbb{R}^k$ represents the representation of a causal variable. Two feed-forward neural networks then output the intervention target vector $\mathcal{I} \in \{0,1\}^d$ and the intervention value vector $S \in \mathbb{R}^d$, with the Gumbel-softmax trick employed for discrete intervention target vector.

## 4.3 TRAINING AND INFERENCE

The overall algorithm is summarized in Algorithm 1. The detals for the prior $p(G|\mathcal{D})$ and $p(\boldsymbol{\theta}|\mathcal{D}, G)$ are given in section 8. Training involves iterative optimization of $\mathcal{R}_T(\pi)$ following Equation 10. We simulate interventional data $\boldsymbol{x}_t \sim p(\boldsymbol{X} \mid G, \boldsymbol{\theta}, \xi_t)$ for samples $(G, \boldsymbol{\theta})$ drawn from the prior. The process involves training the variational posteriors to tighten the lower bound for a fixed policy, followed by improving the policy network via stochastic gradient ascent. At deployment, one only requires a forward pass of the policy network for each experiment, eliminating the need for any intermediate Bayesian inference.

## 5 EXPERIMENTS

In our experiments, we aim to answer two questions empirically: (1) How does our amortized policy network perform across a sequence of interventions compared to methods that select the next best intervention myopically? (2) Can our method generate better interventions for specific causal quantities of interest, as opposed to methods that aim to learn the full causal model? To answer the first question, we compare our method with baselines in the general causal discovery task (Section 5.1). Next, we evaluate its ability to handle targeted causal queries in specific causal reasoning tasks (Section 5.2). A brief overview of the experimental settings is provided below, with further details in Appendix 8.

**Datasets** We evaluate our method on both linear and nonlinear synthetic ground-truth SCMs and realistic gene regulatory network (GRN) datasets. Specifically, we use two types of synthetic datasets

---

**Algorithm 1** The ACED algorithm.

---

1: **Input**: prior $p(G|\mathcal{D})$, $p(\boldsymbol{\theta}|\mathcal{D}, G)$; likelihood $p(\boldsymbol{x}|\mathcal{D}, G, \boldsymbol{\theta}, \boldsymbol{\xi})$, $H$, $p(\psi)$ ; number of stages $T$
2: Initialize policy $\pi$ parameterized with $\gamma$, variational parameters $\boldsymbol{\lambda}$, embedding parameters $\boldsymbol{\phi}$, $\boldsymbol{h}_0 = \{\mathcal{D}\}$ ;
3: **for** $l = 1, \ldots, n_{\text{step}}$ **do**
4:     Simulate $n_{\text{env}}$ samples of graphs $G$, $\boldsymbol{\theta}$, $\psi$ and $\boldsymbol{z}$;
5:     **for** $t = 0, \ldots, T$, **do**
6:         Compute $\boldsymbol{\xi}_t = \pi(\boldsymbol{h}_{t-1})$, then sample $\boldsymbol{x}_t \sim p(\boldsymbol{x}|G, \boldsymbol{\theta}, \boldsymbol{\xi}_t)$;
7:     **end for**
8:     update $\boldsymbol{\gamma}$, $\boldsymbol{\phi}$ and $\boldsymbol{\lambda}$ following gradient ascent, where gradient can be obtained from auto-grad on $\mathcal{R}_T(\pi)$
9: **end for**
10: **Output**: Optimized policy $\pi_{\gamma^*}$; updated variational parameters $\boldsymbol{\lambda}^*$, embedding parameter $\boldsymbol{\phi}^*$

---

generated from Erdős-Rényi (ER) (Erdös & Rényi, 1959) and Scale-Free (SF) graphs with both linear and nonlinear additive noise models (ANM) with various numbers of nodes. Additionally, for the causal discovery task, we use two realistic datasets simulated from DREAM (Greenfield et al., 2010) to evaluate our method's performance on real-world problems. We initialize all cases with $n_{obs} = 50$ observational data $\mathcal{D}$, and we set $n_{int} = 5$ for the number of interventional samples at each stage.

**Baselines**    We compare our method with three baselines in the causal discovery task: **Random-Policy**: Selects both intervention targets and values based on a randomly initialized policy network. **Soft-CBED** (Tigas et al., 2022): Using Bayesian optimization to select intervention targets and values by maximizing EIG at each stage. **DiffCBED** (Tigas et al., 2023): Select intervention targets and values based on a non-adaptive policy network with gradient-based optimization. For the causal reasoning task, we compare our method against the **Random-Policy** and the variational posterior $q_{\boldsymbol{\lambda}}(\boldsymbol{z}|\boldsymbol{h}_T, \mathcal{D})$ is optimized for each policy.

**Metrics**    We use $\log q$ to denote the estimates of the shifted EIG lower bound $\mathcal{R}_T(\pi)$. A higher expectation of $\log q$ corresponds to a higher EIG lower bound, aligning with the problem objective., we use two performance-based metrics to evaluate posterior samples after performing interventions: the expected structural Hamming distance (de Jongh & Druzdzel, 2009) ($\mathbb{E}$-**SHD**) between samples from the posterior model and the ground-truth causal graph, and $F_1$-score for predicting the presence/absence of all edges. We use the learned $q_{\boldsymbol{\lambda}^*}(G|\mathcal{D}, f_{\boldsymbol{\phi}}(\boldsymbol{h}_t))$ for our method and Random-Policy and use DiBS (Lorch et al., 2021), a variational posterior inference method based on Stein variational gradient descent (SVGD) (Liu & Wang, 2016a) for other intervention strategies. For the causal reasoning task, while it is feasible to estimate the shifted EIG on QoI via Nested Monte Carlo (NMC) as proposed in Toth et al. (2022), we first demonstrate an example where NFs could be more efficient for estimating the shifted EIG than NMC. Therefore, we just plot the estimates of the shifted lower bound $\log q$ for reference in synthetic cases.

### 5.1 Experimental Results on Causal Discovery Task

#### 5.1.1 Results on Synthetic Datasets

We evaluate ACED against baselines on both linear and nonlinear SCMs, using Erdös-Rényi (ER) and scale-free (SF) graphs with varying node sizes (10, 20, and 30). We present results for ER graphs in the linear SCM setting and SF graphs in the nonlinear SCM setting. Additional experimental results are provided in Appendix 10.

**Results on Linear synthetic SCMs**    Figure 3 illustrates the performance of different methods across evaluation metrics. In Figure 3(a), ACED achieves a tighter EIG lower bound compared to the random policy, demonstrating that the interventions generated by ACED are likely to more informative in identifying the underlying causal structure. Furthermore, as shown in Figures 3(b) and (c), ACED significantly reduces the expected structural Hamming distance ($\mathbb{E}$-SHD) and improves the $F_1$-score compared to other methods. These improvements highlight the effectiveness of ACED in designing interventions that improve the identifiability of the true causal structure. Notably, while

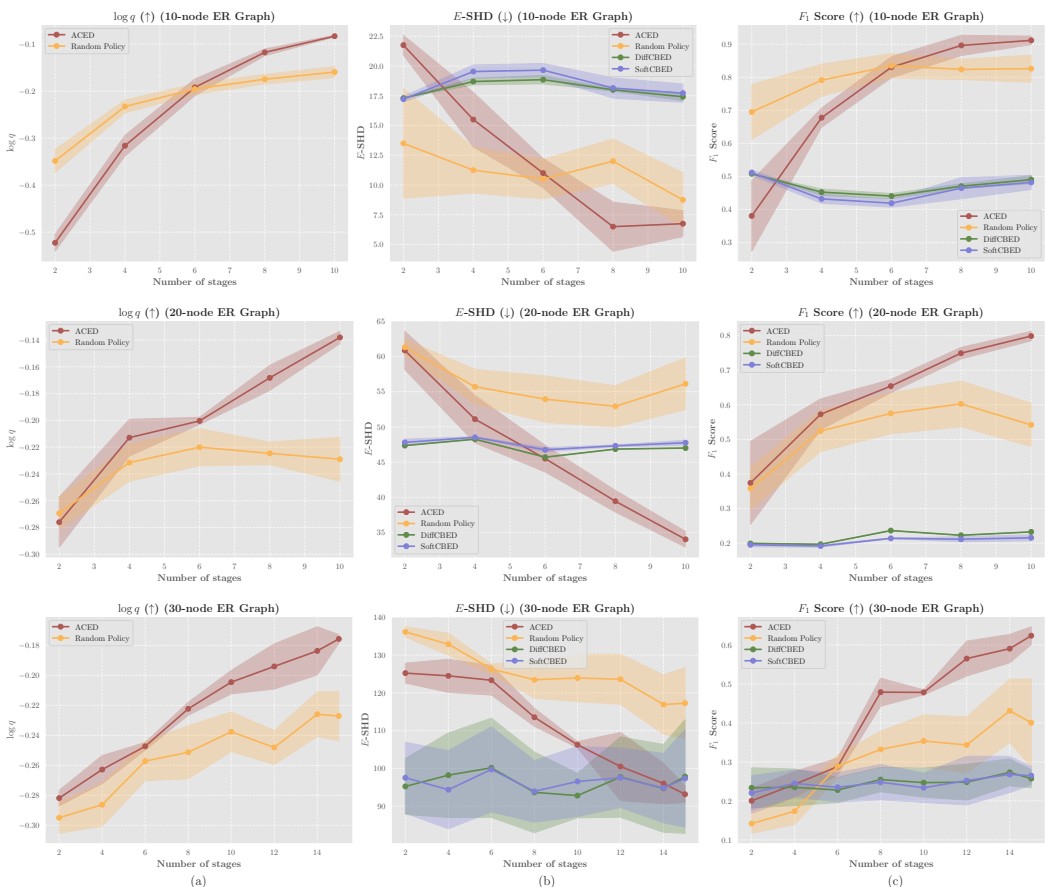

Figure 3: Results of different causal experimental design methods on 10, 20, and 30-node Erdös-Rényi (ER) graphs with linear additive noise models. ACED achieves a tighter EIG lower bound, significantly reducing $\mathbb{E}$-SHD and improving the $F_1$-score compared to the baselines, especially as the number of stages increases.

ACED initially performs worse than the baselines, it surpasses them as the number of stages increases. This suggests that ACED focuses on designing interventions that are globally optimal across multiple stages, rather than optimizing the next-step intervention. Although some baselines achieve a low $\mathbb{E}$-SHD as the number of nodes increases, this is likely due to the posterior inference model DiBS converging to low-entropy solutions, which tend to predict only a few edges. When considering both $\mathbb{E}$-SHD and $F_1$-score, ACED outperforms the baselines significantly.

**Results on Nonlinear synthetic SCMs** We next consider the more challenging setting of nonlinear SCMs, where SF graphs are used due to their more informative prior compared to ER graphs. Figure 4 also demonstrates that ACED consistently outperforms the baselines across all metrics. However, the improvements become marginal compared to the linear SCM setting, especially with the random policy. Furthermore, results on ER graphs are presented in Figure 10 in the Appendix, where the random policy achieves performance comparable to ACED. These findings suggest that more extensive training data and more informative priors may be required to improve the training and amortization of the policy network in complex nonlinear settings.

### 5.1.2 RESULTS ON REALISTIC GRN DATASETS

In addition to synthetic datasets, we consider two more realistic GRN datasets, Yeast and Ecoli with 10 nodes, and the results are presented in Table 1. ACED's performance gap widens further on these realistic datasets, demonstrating its effectiveness when the prior is more informative. On the Ecoli dataset, ACED achieves near-perfect performance across all metrics, significantly outperforming

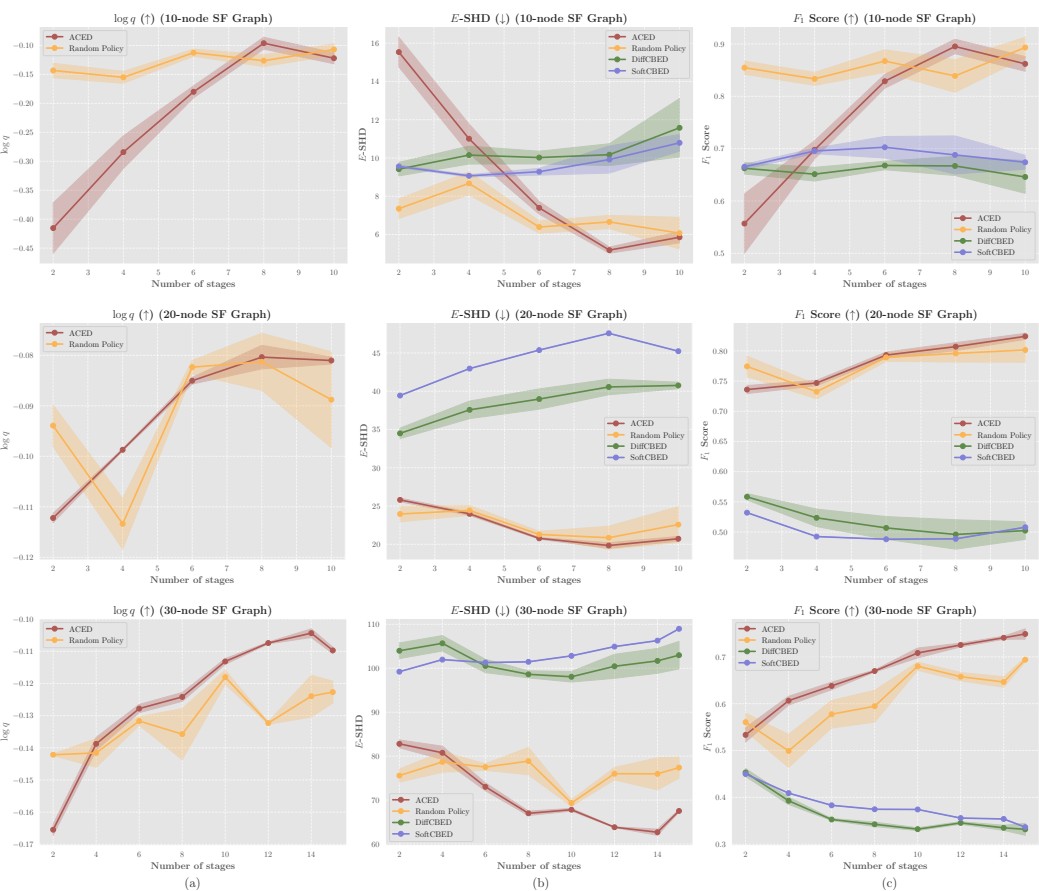

Figure 4: Results of different causal experimental design methods on 10, 20, and 30-node scale-free (SF) graphs with nonlinear additive noise models. ACED maintains a tighter EIG lower bound and superior performance on $\mathbb{E}$-SHD and $F_1$-score, though improvements are less consistent in this more challenging nonlinear setting.

all baselines. Interestingly, the Random-Policy baseline performs surprisingly well on the Yeast dataset, suggesting that this particular graph structure might be easier to learn. However, ACED still outperforms it consistently across all metrics.

## 5.2 EXPERIMENTAL RESULTS ON CAUSAL REASONING TASK

For causal reasoning, we first illustrate that though it is feasible to find the optimal policy on general QoI via Nested Monte Carlo (NMC) estimator as explained in Appendix 7.1, NMC might have high bias and variance compared with estimating EIG lower bound via Normalizing Flows. Consider a fixed graph with 4 nodes, where QoI is the effect on node 3 and node 4 under the fixed do operation $\mathrm{do}(X_2 = 2)$, and the node for intervention is

|  | Yeast-10 | | Ecoli-10 | |
|---|---|---|---|---|
| Method | SHD ($\downarrow$) | $F_1$-score ($\uparrow$) | SHD ($\downarrow$) | $F_1$-score ($\uparrow$) |
| Random-Policy | 4.30±4.80 | 0.896 ±0.20 | 15.8±5.78 | 0.403±0.29 |
| Soft-CBED | 9.44±7.38 | 0.578±0.44 | 20.62±4.38 | 0.184±0.22 |
| DiffCBED | 8.78±9.08 | 0.784±0.25 | 18.37±2.99 | 0.298±0.20 |
| ACED | **1.90±3.59** | **0.916±0.16** | **1.10±0.83** | **0.979±0.06** |

Table 1: Performance comparison on 10-node Yeast and Ecoli Gene Regulatory Networks. Values show mean ± standard deviation over 10 random seeds. Best results in bold.

node 1. The overall setup is given in Figure 5, where all observations are associated with noise $\epsilon_j \sim N(0, 0.3^2)$.

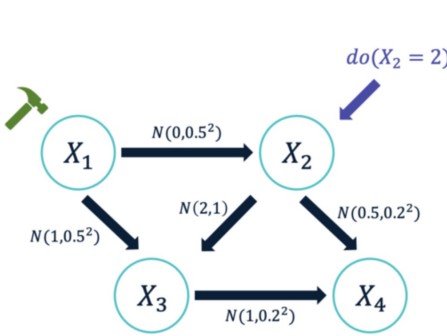

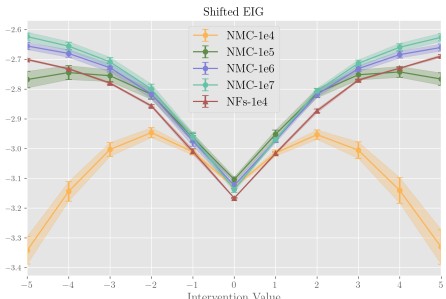

Figure 5: Intervetion on node 1 with integers from [-5, -4, ..., 5]. QoIs are node 3 and 4 under the intervention $\text{do}(X_2 = 2)$.

Figure 6: Shifted EIG estimates from NFs and NMC. The lines represent the mean of 4 initializations of $\boldsymbol{\lambda}$ for NFs and 4 replicates with different random seeds for NMC, with the shaded areas indicating one standard error.

Considering uniform interventions on node 1 with values [-5, -4, ..., 5], the corresponding shifted EIG estimated with NMC under outer and inner sample size being 10000, 20000 40000, respectively. For NFs, we optimize the EIG lower bound with respect to $\boldsymbol{\lambda}$ using 20000 samples. The comparison is shown in Figure 6.

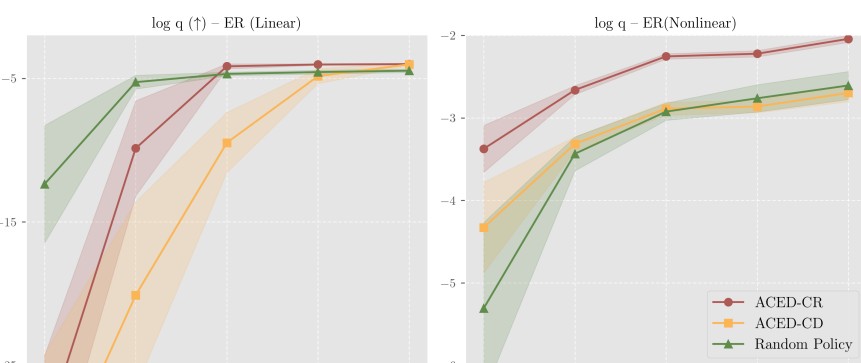

Figure 7: Shifted EIG lower bound for QoI on an Erdos Renyi graph with 10 nodes. The left figure plots the results on a linear mechanism with $[X_1, X_6]^{\text{do}(X1=\psi)}, \psi \sim N(3, 0.5^2)$. The right plots the results on an non-linear mechanism with $[X_3, X_9]^{\text{do}(X3=\psi)}, \psi \sim N(3, 0.5^2)$. Mean and standard deviation (shaded) of 4 replicates with different random seeds for initializing parameters, each replicate evaluated on 500 graphs.

From Figure 6, NMC exhibits a large bias and variance especially near the boundary values, and while increasing the sample size for NMC leads to smaller variance and higher estimates at the boundaries, NMC still struggles to identify the optimal interventions at these points. While NFs theoretically provide a lower bound estimator for EIG, the NFs estimates are only upper bounded by NMC estimates in the middle range, where NMC shows low bias and variance. Remarkably, NFs not only identify the optimal EIG at the boundary with far fewer samples but also surpasses all NMC estimates at the boundaries. This suggests that NFs could be significantly more efficient in identifying the optimal design, even in this simple case with a small number of nodes and a fixed graph. Thus, for the remaining comparisons, we focus on plotting the NFs estimates for the EIG lower bound.

The more complicated causal reasoning tasks are implemented on an Erdös-Rényi graph with 10 nodes, under both linear and nonlinear mechanisms, and the results are plotted in Figure 7. In the

linear model, the policy trained for causal discovery performs comparably to the policy optimized for reasoning. This is potentially because the uncertainty on graph has been effectively reduced by the policy and the relationship between graph and QoI is relatively straightforward. However, in the nonlinear case, the policy trained for causal reasoning significantly outperforms other policies, demonstrating the advantage of a tailored policy when the causal mechanism is complex.

# 6 DISCUSSION

In this paper, we introduced Adaptive Causal Experimental Design (ACED), a novel approach to Bayesian optimal experimental design for flexible causal queries. ACED addresses three key limitations of existing methods: (1) myopic design–by learning a policy considering future experiments; (2) high computational costs at deployment–by training an amortized design policy network, allowing rapid decisions given interventional data at each stage; and (3) inefficiency in learning the full causal model for specific queries–by targeting flexible QoIs instead of focusing on learning the entire causal graph. Our theoretical contributions include deriving variational lower bounds for policy EIG on general causal queries and developing a framework for learning non-myopic, adaptive strategies. Empirical evaluations on both synthetic and real-world-inspired datasets demonstrated ACED's superior performance across various graph sizes and structures, consistently outperforming existing baselines in terms of accuracy and computational efficiency.

**Limitations and future work**   While ACED demonstrates significant effectiveness, several limitations and opportunities for future research remain. While the independent Bernoulli distribution can produce high-quality posterior graphs, its inherent independence assumption introduces bias in EIG estimation. Besides, the probability of producing cyclic graphs increases as the number of nodes approaches to 30, which hampers the overall sampling process and motivates a careful incorporation of acyclic constraints. Moreover, while the policy has theoretical guarantees, its robustness against changes in the design horizon—such as budget cuts or extensions—remains unclear. Similarly, the need to adjust the model structure midway through the intervention stages also requires further exploration and testing. Further efforts could also focus on extending this framework to multi-target settings, where interventions can be jointly performed on multiple nodes at each stage.

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

# 7 APPENDIX

## 7.1 PROOFS

**Total EIG for policy**    The EIG of a policy over a sequence of experiments on QoI is given as

$$
\begin{aligned}
\mathcal{I}_T(\pi) &= \mathbb{E}_{p(\mathcal{M}|\mathcal{D})p(\boldsymbol{h}_T|\mathcal{D},\mathcal{M},\pi)}[\sum_{t=1}^{T}\mathcal{I}_t(\boldsymbol{\xi}_t)] \\
&= \mathbb{E}_{p(\mathcal{M}|\mathcal{D})p(\boldsymbol{h}_T|\mathcal{D},\mathcal{M},\pi)}\left[\sum_{t=1}^{T}\mathbb{E}_{p(\boldsymbol{x}_t|\boldsymbol{\xi}_t)}\left[\mathbb{D}_{KL}[p(\boldsymbol{z}|\mathcal{D},\boldsymbol{h}_{t-1},\boldsymbol{x}_t,\boldsymbol{\xi}_t)||p(\boldsymbol{z}|\mathcal{D},\boldsymbol{h}_{t-1})]\right]\right] \\
&= \mathbb{E}_{p(\mathcal{M}|\mathcal{D})p(\boldsymbol{h}_T|\mathcal{D},\mathcal{M},\pi)}\left[\sum_{t=1}^{T}\mathbb{E}_{p(\boldsymbol{x}_t|\boldsymbol{\xi}_t)}\left[\mathbb{E}_{p(\boldsymbol{z}|\mathcal{D},\boldsymbol{h}_{t-1},\boldsymbol{x}_t,\boldsymbol{\xi}_t)}\log[\frac{p(\boldsymbol{z}|\mathcal{D},\boldsymbol{h}_{t-1},\boldsymbol{x}_t,\boldsymbol{\xi}_t)}{p(\boldsymbol{z}|\mathcal{D},\boldsymbol{h}_{t-1})}]\right]\right] \\
&= \sum_{t=1}^{T}\left[\mathbb{E}_{p(\mathcal{M}|\mathcal{D})}\mathbb{E}_{p(\boldsymbol{h}_t|\mathcal{D},\mathcal{M},\pi)}\mathbb{E}_{p(\boldsymbol{x}_t|\boldsymbol{\xi}_t)}\left[\mathbb{E}_{p(\boldsymbol{z}|\mathcal{D},\boldsymbol{h}_{t-1},\boldsymbol{x}_t,\boldsymbol{\xi}_t)}\log[\frac{p(\boldsymbol{z}|\mathcal{D},\boldsymbol{h}_{t-1},\boldsymbol{x}_t,\boldsymbol{\xi}_t)}{p(\boldsymbol{z}|\mathcal{D},\boldsymbol{h}_{t-1})}]\right]\right]
\end{aligned}
\tag{12}
$$

The outer expectation over $\mathcal{M}$ is independent to the inner terms and thus could be marginalized out. Additionally, for $t_i > t_j$, $\boldsymbol{h}_{t_j:t_i}$ is independent to the $\boldsymbol{h}_{t_j}$ in the inner log term and could be also marginalized out. Lastly, notice $\boldsymbol{h}_t = \{\boldsymbol{x}_t, \boldsymbol{\xi}_t\}$, then we can write eq. (12) as

$$
\begin{aligned}
\mathcal{I}_T(\pi) &= \sum_{t=1}^{T}\left[\mathbb{E}_{p(\boldsymbol{h}_t|\mathcal{D},\pi)}\left[\mathbb{E}_{p(\boldsymbol{z}|\mathcal{D},\boldsymbol{h}_t)}\log[\frac{p(\boldsymbol{z}|\mathcal{D},\boldsymbol{h}_t)}{p(\boldsymbol{z}|\mathcal{D},\boldsymbol{h}_{t-1})}]\right]\right] \\
&= \sum_{t=1}^{T}\left[\mathbb{E}_{p(\boldsymbol{h}_t,\boldsymbol{z}|\mathcal{D},\pi)}\log[\frac{p(\boldsymbol{z}|\mathcal{D},\boldsymbol{h}_t)}{p(\boldsymbol{z}|\mathcal{D},\boldsymbol{h}_{t-1})}]\right] \\
&= \sum_{t=1}^{T}\left[\mathbb{E}_{p(\boldsymbol{h}_t,\boldsymbol{z}|\mathcal{D},\pi)}\log p(\boldsymbol{z}|\mathcal{D},\boldsymbol{h}_t) - \mathbb{E}_{p(\boldsymbol{h}_{t-1},\boldsymbol{z}|\mathcal{D})}\log[p(\boldsymbol{z}|\mathcal{D},\boldsymbol{h}_{t-1})]\right] \\
&= \mathbb{E}_{p(\boldsymbol{h}_T,\boldsymbol{z}|\mathcal{D},\pi)}[\log p(\boldsymbol{z}|\mathcal{D},\boldsymbol{h}_T)] - \mathbb{E}_{p(\boldsymbol{h}_0,\boldsymbol{z}|\mathcal{D})}\log[p(\boldsymbol{z}|\mathcal{D},\boldsymbol{h}_0)] \\
&= \mathbb{E}_{p(\boldsymbol{h}_T,\boldsymbol{z}|\mathcal{D},\pi)}[\log p(\boldsymbol{z}|\mathcal{D},\boldsymbol{h}_T)] - \mathbb{E}_{p(\boldsymbol{z}|\mathcal{D})}[\log p(\boldsymbol{z}|\mathcal{D})] \\
&= \mathbb{E}_{p(\mathcal{M}|\mathcal{D})}\mathbb{E}_{p(\boldsymbol{h}_T|\mathcal{D},\mathcal{M},\pi)}\mathbb{E}_{p(\boldsymbol{z}|\mathcal{D},\mathcal{M},\pi)}[\log p(\boldsymbol{z}|\mathcal{D},\boldsymbol{h}_T)] \\
&\qquad\qquad - \mathbb{E}_{p(\mathcal{M}|\mathcal{D})}\mathbb{E}_{p(\boldsymbol{h}_T|\mathcal{D},\mathcal{M},\pi)}\mathbb{E}_{p(\boldsymbol{z}|\mathcal{D})}[\log p(\boldsymbol{z}|\mathcal{D})] \\
&= \mathbb{E}_{p(\mathcal{M}|\mathcal{D})p(\boldsymbol{h}_T|\mathcal{D},\mathcal{M},\pi)p(\boldsymbol{z}|\mathcal{D},\mathcal{M})}\left[\log\frac{p(\boldsymbol{z}|\mathcal{D},\boldsymbol{h}_T)}{p(\boldsymbol{z}|\mathcal{D})}\right]
\end{aligned}
\tag{13}
$$

where $\boldsymbol{h}_0 = \{\mathcal{D}\}$ if $\mathcal{D}$ is available otherwise $\boldsymbol{h}_0 = \emptyset$.

**Nested Monte Carlo on shifted EIG** To estimate the EIG of a policy on QoIs, since the denominator term is independent to $\pi$, a shifted EIG is derived as

$$
\begin{aligned}
\mathcal{I}_T(\pi) &= \mathbb{E}_{p(G,\boldsymbol{\theta}|\mathcal{D})p(\boldsymbol{h}_T|\mathcal{D},G,\boldsymbol{\theta},\pi)p(\boldsymbol{z}|\mathcal{D},G,\boldsymbol{\theta})}\left[\log\frac{p(\boldsymbol{z}|\mathcal{D},\boldsymbol{h}_T)}{p(\boldsymbol{z}|\mathcal{D})}\right] \\
&= \mathbb{E}_{p(\mathcal{M}|\mathcal{D})p(\boldsymbol{h}_T|\mathcal{D},\mathcal{M},\pi)p(\boldsymbol{z}|\mathcal{D},\mathcal{M})}\left[\log\frac{p(\boldsymbol{z}|\mathcal{D},\boldsymbol{h}_T)}{p(\boldsymbol{z}|\mathcal{D})}\right] \\
&= \mathbb{E}_{p(\mathcal{M}|\mathcal{D})p(\boldsymbol{h}_T|\mathcal{D},\mathcal{M},\pi)p(\boldsymbol{z}|\mathcal{D},\mathcal{M})}\left[\log p(\boldsymbol{z}|\mathcal{D},\pi,\boldsymbol{h}_T)\right] - c \\
&= \mathbb{E}_{p(\mathcal{M}|\mathcal{D})p(\boldsymbol{h}_T|\mathcal{D},\mathcal{M},\pi)p(\boldsymbol{z}|\mathcal{D},\mathcal{M})}\left[\log\frac{p(\boldsymbol{z},\boldsymbol{h}_T|\mathcal{D},\pi)}{p(\boldsymbol{h}_T|\mathcal{D},\pi)}\right] - c \\
&= \mathbb{E}_{p(\mathcal{M}|\mathcal{D})p(\boldsymbol{h}_T,\boldsymbol{z}|\mathcal{D},\mathcal{M},\pi)}[\log p(\boldsymbol{z},\boldsymbol{h}_T|\mathcal{D},\pi)] \\
&\quad - \mathbb{E}_{p(\mathcal{M}|\mathcal{D})p(\boldsymbol{h}_T,\boldsymbol{z}|\mathcal{D},\mathcal{M},\pi)}[\log p(\boldsymbol{h}_T|\mathcal{D},\pi)] - c
\end{aligned}
\tag{14}
$$

where $c = \mathbb{E}_{p(\mathcal{M}|\mathcal{D})p(\boldsymbol{z}|\mathcal{D},\mathcal{M})}[\log p(\boldsymbol{z}|\mathcal{D})]$ is a constant with respect to the policy network. Therefore,

$$
\begin{aligned}
\arg\max_{\pi}\mathcal{I}_T(\pi) = \arg\max_{\pi}\bigg[ &-\mathbb{E}_{p(\mathcal{M}|\mathcal{D})p(\boldsymbol{h}_T|\mathcal{D},\mathcal{M},\pi)}\log[p(\boldsymbol{h}_T|\mathcal{D},\pi)] \\
&+ \mathbb{E}_{p(\mathcal{M}|\mathcal{D})}[\mathbb{E}_{p(\boldsymbol{z},\boldsymbol{h}_T|\mathcal{M},\mathcal{D},\pi)}[\log\mathbb{E}_{p(\mathcal{M}'|\mathcal{D})}[p(\boldsymbol{h}_T,\boldsymbol{z}|\mathcal{M}',\mathcal{D})]]]\bigg]
\end{aligned}
\tag{15}
$$

The right part in eq. (15) can be estimated via nested Monte Carlo (NMC) estimator via

$$
\begin{aligned}
&-\mathbb{E}_{p(\mathcal{M}|\mathcal{D})p(\boldsymbol{h}_T|\mathcal{D},\mathcal{M},\pi)}\log[p(\boldsymbol{h}_T|\mathcal{D},\pi)] + \mathbb{E}_{p(\mathcal{M}|\mathcal{D})}[\mathbb{E}_{p(\boldsymbol{z},\boldsymbol{h}_T|\mathcal{M},\mathcal{D},\pi)}[\log[\mathbb{E}_{p(\mathcal{M}'|\mathcal{D})}p(\boldsymbol{h}_T,\boldsymbol{z}|\mathcal{M}',\mathcal{D})]]] \\
&\approx -\frac{1}{N}\sum_{i=1}^{N}\log\frac{1}{M}\sum_{j=1}^{M}p(\boldsymbol{h}_T^i|\mathcal{D},\mathcal{M}'^j,\pi) + \frac{1}{N}\sum_{i=1}^{N}\log\frac{1}{M}\sum_{j=1}^{M}p(\boldsymbol{h}_T^i,\boldsymbol{z}^i|\mathcal{M}'^j)
\end{aligned}
$$

where $\boldsymbol{h}_T^i, \boldsymbol{z}^i$ is simulated from $p(\boldsymbol{h}_T,\boldsymbol{z}|\pi,\mathcal{M}^i)$, and $\mathcal{M}^i, \mathcal{M}'^j \sim p(\mathcal{M})$).

**Variational EIG lower bound for Policy** $\mathcal{I}_{T;L}(\pi)$ can be shown to be a lower bound of $\mathcal{I}_T(\pi)$ since

$$
\begin{aligned}
\mathcal{I}_T(\pi) - \mathcal{I}_{T;L}(\pi) &= \mathbb{E}_{p(\mathcal{M}|\mathcal{D})p(\boldsymbol{h}_T|\mathcal{D},\mathcal{M},\pi)p(\boldsymbol{z}|\mathcal{D},\mathcal{M})}\left[\log\frac{p(\boldsymbol{z}|\mathcal{D},\boldsymbol{h}_T)}{p(\boldsymbol{z}|\mathcal{D})}\right] \\
&\quad - \mathbb{E}_{p(\mathcal{M}|\mathcal{D})p(\boldsymbol{h}_T|\mathcal{D},\mathcal{M},\pi)p(\boldsymbol{z}|\mathcal{D},\mathcal{M})}\left[\log\frac{q_{\boldsymbol{\lambda}}(\boldsymbol{z}|\mathcal{D},f_{\boldsymbol{\phi}}(\boldsymbol{h}_T))}{p(\boldsymbol{z}|\mathcal{D})}\right] \\
&= \mathbb{E}_{p(\mathcal{M}|\mathcal{D})p(\boldsymbol{h}_T|\mathcal{D},\mathcal{M},\pi)p(\boldsymbol{z}|\mathcal{D},\mathcal{M})}\left[\log\frac{p(\boldsymbol{z}|\mathcal{D},\boldsymbol{h}_T)}{q_{\boldsymbol{\lambda}}(\boldsymbol{z}|\mathcal{D},f_{\boldsymbol{\phi}}(\boldsymbol{h}_T))}\right] \\
&= \mathbb{E}_{p(\mathcal{M}|\mathcal{D})p(\boldsymbol{h}_T|\mathcal{D},\mathcal{M},\pi)}\left[\mathbb{D}_{KL}(p(\boldsymbol{z}|\mathcal{D},\boldsymbol{h}_T)\,||\,q_{\boldsymbol{\lambda}}(\boldsymbol{z}|\mathcal{D},f_{\boldsymbol{\phi}}(\boldsymbol{h}_T)))\right]
\end{aligned}
\tag{16}
$$

This is non-negative as the KL-divergence is non-negative, and the lower bound is tight if and only if $q_{\boldsymbol{\lambda}}(\boldsymbol{z}|\mathcal{D},f_{\boldsymbol{\phi}}(\boldsymbol{h}_T)) = p(\boldsymbol{z}|\mathcal{D},\boldsymbol{h}_T)$ for all $(\boldsymbol{z},\boldsymbol{h}_T)$ pairs under the policy. Notice eq. (16) can also results from the difference between the RHS in eq. (15) and $\mathcal{R}_T(\pi)$, indicating another upper bound relationship.

## 7.2 NORMALIZING FLOWS FOR CAUSAL REASONING

An NF is an invertible mapping from a target random variable $\boldsymbol{Z}$ to a standard normal variable $\boldsymbol{\eta}$, $\boldsymbol{Z} = g(\boldsymbol{\eta})$ (and $\boldsymbol{\eta} = f(\boldsymbol{Z})$ where $f = g^{-1}$), via a composition of successive invertible mappings. The PDFs between these variables are related via

$$p(\boldsymbol{z}) = p_{\boldsymbol{\eta}}(f(\boldsymbol{z}))|\det\frac{\partial f(\boldsymbol{z})}{\partial \boldsymbol{z}}| \tag{17}$$

Writing in a successive mapping form $\boldsymbol{z} = g(\boldsymbol{\eta}) = g_1 \circ g_2 \circ ... \circ g_n(\boldsymbol{\eta}) = g_1(g_2(...(g_n(\boldsymbol{\eta}))...))$ with $n \geq 1$ invertible transformations, the log density is

$$\log p(\boldsymbol{z}) = \log p_{\boldsymbol{\eta}}(f_n \circ f_{n-1} \circ ... \circ f_1(\boldsymbol{z})) + \sum_{i=1}^{n} \log |\det\frac{\partial f_i \circ f_{i-1} \circ ... f_1(\boldsymbol{z})}{\partial \boldsymbol{z}}| \tag{18}$$

where $\boldsymbol{\eta} = f(\boldsymbol{z}) = f_n \circ f_{n-1} \circ ... \circ f_1(\boldsymbol{z})$ and $f_i = g_i^{-1}$. The successive transformations on $\boldsymbol{\eta}$ can achieve a highly expressive density for the target variable $\boldsymbol{Z}$ Dinh et al. (2016).

To approximate the QoI posterior $q_{\boldsymbol{\lambda}}(\boldsymbol{z}|\mathcal{D}, f_{\boldsymbol{\phi}}(\boldsymbol{h}_T))$, we use compositions of successive coupling layers, which partitions $\boldsymbol{z}$ into two parts $\boldsymbol{z} = [\boldsymbol{z}_1, \boldsymbol{z}_2]^T$ in similar dimensions $n_{\boldsymbol{z}_1}, n_{\boldsymbol{z}_2}$, and introduces invertible mappings in the form as:

$$f_1(\boldsymbol{z}) = \begin{pmatrix} \boldsymbol{z}_1 \\ \tilde{\boldsymbol{z}}_2 = \boldsymbol{z}_2 \odot \exp(s_1(\boldsymbol{z}_1)) + t_1(\boldsymbol{z}_1) \end{pmatrix}$$

$$f_2(f_1(\boldsymbol{z})) = \begin{pmatrix} \tilde{\boldsymbol{z}}_1 = \boldsymbol{z}_1 \odot \exp(s_2(\tilde{\boldsymbol{z}}_2)) + t_2(\tilde{\boldsymbol{z}}_2) \\ \tilde{\boldsymbol{z}}_2 \end{pmatrix} \tag{19}$$

where $s_1, t_1$ map $\mathbb{R}^{n_{\boldsymbol{z}_1}} \mapsto \mathbb{R}^{n_{\boldsymbol{z}_2}}$ and $s_2, t_2$ map $\mathbb{R}^{n_{\boldsymbol{z}_2}} \mapsto \mathbb{R}^{n_{\boldsymbol{z}_1}}$, and $\odot$ denotes element-wise product. The Jacobian of $f_1$ is

$$\begin{bmatrix} \mathbb{I}_d & 0 \\ \frac{\partial f_1(\boldsymbol{z})}{\partial \boldsymbol{z}_2} & \text{diag}(\exp[s_1(\boldsymbol{z}_1)]) \end{bmatrix},$$

a lower triangular matrix with determinant $\exp[\sum_{j=1}^{n_{\boldsymbol{z}_2}} s_1(\boldsymbol{z}_1)_j]$. Similarly the Jacobian of $f_2$ is an upper triangular matrix with determinant $\exp[\sum_{j=1}^{n_{\boldsymbol{z}_1}} s_2(\tilde{\boldsymbol{z}}_2)_j]$. $s$'s and $t$'s can be represented via, for example, NNs for their expressiveness. Multiple such transformations ($n_{\text{trans}}$) from eq. (19) can be composed to further increase expressiveness of the overall mapping. To incorporate the dependency of posterior on $\boldsymbol{h}_T$, the $s(\cdot)$ and $t(\cdot)$ are set to additionally take $f_{\boldsymbol{\phi}(\boldsymbol{h}_T)}$ as input.

# 8 EXPERIMENT DETAILS

In the synthetic cases, we generate the Erdös-Rényi and Scale Free. In the semi-synthetic setting, we use the two real-world inspired gene regulatory networks based on Greenfield et al. (2010).

**Erdös-Rényi** For Erdös-Rényi, each edge is sampled independently with a fixed probability $p$. Given $n$ nodes, it generates a graph where the number of edges follows a binomial distribution, with the expected number of edges being $p \times \binom{n}{2}$. We follow Lorch et al. (2022) and scale this probability to obtain $O(d)$ edges in expectation. We use NetworkX Hagberg et al. (2008) and method `fast_gnp_random_graph` Batagelj & Brandes (2005) to generate Erdös-Rényi graphs.

**Scale Free** A Scale-Free graph is characterized by a power-law degree distribution Barabási & Albert (1999). In a scale free graph, a small number of nodes have a relatively large number of connections, while most nodes have relatively few connections.

**Realstic Gene Regulatory Networks** In the semi-synthetic setting, we utilize the DREAM (Dialogue for Reverse Engineering Assessments and Methods) benchmarks Greenfield et al. (2010), which are specifically designed to evaluate computational methods for reverse engineering biological networks. DREAM provides realistic simulations of gene regulatory and protein signaling networks, generated through GeneNetWeaver v3.12. This simulator employs both ordinary differential equations (ODEs) and stochastic differential equations (SDEs) to accurately model complex biological mechanisms and their inherent noise. In our experiments, we focus on two specific subnetworks from the DREAM benchmark: the E. coli and Yeast networks, each consisting of 10 nodes representing the true causal graph, and simulate the mechanisms following the setup described in Tigas et al. (2023).

**Linear additive model**   In the linear domain, we model the parent-child relationship via

$$x_j = \boldsymbol{\theta}_j^T \boldsymbol{x}_{par_j} + \epsilon \qquad \epsilon \sim N(0, \sigma^2) \tag{20}$$

We assume $\theta \sim N(0, 2)$ in the prior and $\sigma^2 = 0.1$.

**Nonlinear additive model**   In the nonlinear model, we model the parent-child relationship via FFN with 2 hidden layers and ReLU activation. We assume a standard normal prior for neural network weights and biases.

**Posterior inference with observational data**   For prior $p(G|\mathcal{D})$ and $p(\boldsymbol{\theta}|\mathcal{D}, G)$, we first initialize a set of $\mathcal{D}$ from the ground truth graph for all synthetic cases. While the other benchmarks perform inference using Dibs Lorch et al. (2021), a SVGD based inference algorithm. We perform Bayesian inference in two steps: first we infer the graph structure $G$, we follow Lorch et al. (2022) and train an amortized $q_{\boldsymbol{\lambda}}(f_{\boldsymbol{\phi}}(\mathcal{D}))$ by minimizing

$$\mathbb{E}_{p(\mathcal{D})}\left[\mathbb{D}_{KL}(p(G|\mathcal{D})||q_{\boldsymbol{\lambda}}(f_{\boldsymbol{\phi}}(\mathcal{D})))\right] \tag{21}$$

with respect to $\boldsymbol{\lambda}$ and $\boldsymbol{\phi}$, where an independent Bernoulli distribution for each edge is adopted for $q_{\boldsymbol{\lambda}}(\cdot)$.

Once we learned the posterior over the graph, we obtained a column of $p :, j$ representing the probabilities of causal edges from other nodes to node $j$. For each $j$, we simulate 200 realization from $p :, j$, yielding potential parent sets for node $j$, with the probability of each parent set weighted by its appearance frequency.

For inference on $\boldsymbol{\theta}$, let $\boldsymbol{\theta}_j$ denote the parameters involved in the relationship between $X_{par(j)}$ to $X_j$, then the overall posterior can be factorized as:

$$p(\boldsymbol{\theta}|\boldsymbol{x}, \mathcal{D}, G) = \prod_j p(\boldsymbol{\theta}_j|\boldsymbol{x}_{par(j)}, \boldsymbol{x}_j, \mathcal{D}, G) \tag{22}$$

with a proof given below:

$$p(\boldsymbol{\theta}|\boldsymbol{x}, \mathcal{D}, G) \propto \prod_j p(\boldsymbol{\theta}_j|\mathcal{D}, G)p(\boldsymbol{x}_j|\boldsymbol{x}_{par(j)}, \boldsymbol{\theta}_j, \mathcal{D}, G)$$

whereas the right hand side in eq. (22) can be also expressed as:

$$\prod_j p(\boldsymbol{\theta}_j|\boldsymbol{x}_{par(j)}, \boldsymbol{x}_j, \mathcal{D}, G) = \prod_j \frac{p(\boldsymbol{\theta}_j, \boldsymbol{x}_{par(j)}, \boldsymbol{x}_j, \mathcal{D}, G)}{p(\boldsymbol{x}_j, \boldsymbol{x}_{par(j)}, \mathcal{D}, G)} \tag{23}$$

$$= \prod_j \frac{p(\boldsymbol{\theta}_j, \boldsymbol{x}_{par(j)}, \boldsymbol{x}_j|\mathcal{D}, G)p(\mathcal{D}, G)}{p(\boldsymbol{x}_j, \boldsymbol{x}_{par(j)}|\mathcal{D}, G)p(\mathcal{D}, G)}$$

$$= \prod_j \frac{p(\boldsymbol{\theta}_j, \boldsymbol{x}_{par(j)}, \boldsymbol{x}_j|\mathcal{D}, G)}{p(\boldsymbol{x}_j, \boldsymbol{x}_{par(j)}|\mathcal{D}, G)}$$

$$= \prod_j \frac{p(\boldsymbol{\theta}_j|\mathcal{D}, G)p(\boldsymbol{x}_{par(j)}, \boldsymbol{x}_j|\boldsymbol{\theta}_j, \mathcal{D}, G)}{p(\boldsymbol{x}_j|\boldsymbol{x}_{par(j)}, \mathcal{D}, G)p(\boldsymbol{x}_{par(j)}|\mathcal{D}, G)}$$

$$= \prod_j \frac{p(\boldsymbol{\theta}_j|\mathcal{D}, G)p(\boldsymbol{x}_j|\boldsymbol{x}_{par(j)}, \boldsymbol{\theta}_j, \mathcal{D}, G)p(\boldsymbol{x}_{par(j)}|\theta_{:,j}, \mathcal{D}, G)}{p(\boldsymbol{x}_j|\boldsymbol{x}_{par(j)}, \mathcal{D}, G)p(\boldsymbol{x}_{par(j)}|\mathcal{D}, G)}$$

$$\propto \prod_j p(\boldsymbol{\theta}_j|\mathcal{D}, G)p(\boldsymbol{x}_j|\boldsymbol{x}_{par(j)}, \boldsymbol{\theta}_j, \mathcal{D}, G)$$

the last equation follows as $\boldsymbol{x}_{par(j)}$ is independent of $\boldsymbol{\theta}_j$ and the denominator is a constant with respect to $\boldsymbol{\theta}$.

Therefore, for each unique parent set of $X_j$, we perform inference on $\boldsymbol{\theta}_j$ using observed $\boldsymbol{x}_{par(j)}$ and $\boldsymbol{x}_j$ independently for each $j$. For linear models, we run a MCMC Hoffman et al. (2014) and

store 400 posterior samples of $\theta_j$. During the sampling in eq. (9), for $p(G|\mathcal{D})$, we draw potential parents set for each node $j$ and form a DAG, then we draw $p(\boldsymbol{\theta_j}|G, \mathcal{D})$ from the stored MCMC samples corresponding to the respective parent structures. Among the 400 posterior samples, 80% are potentially drawn during training, and evaluation is performed on the samples in the remaining 20%. For nonlinear neural network, we apply Pyro's Stochastic Variational Inference (SVI) Bingham et al. (2019) to obtain a mean-field Gaussian approximation to the true posterior $p(\boldsymbol{\theta}_j|G, \mathcal{D})$.

## 9 Further Discussion of Related Work

**(Bayesian) Causal Discovery**   Causal discovery has been widely studied in machine learning and statistics (Glymour et al., 2019; Heinze-Deml et al., 2018; Peters et al., 2017; Vowels et al., 2022). In contrast to traditional causal discovery approaches infer a single causal graph from observational data (Brouillard et al., 2020; Hauser & Bühlmann, 2012; Lippe et al., 2021; Perry et al., 2022; Peters et al., 2016; Heinze-Deml et al., 2018), Bayesian causal discovery (Friedman & Koller, 2003; Heckerman et al., 2006; Tong & Koller, 2001) aims to infer a posterior distribution over SCMs and their DAGs from observed data. Recent works (Cundy et al., 2021; Lorch et al., 2021; Annadani et al., 2021) proposed a variational approximation of the posterior over the DAGs which allowed for modeling a distribution rather than a point estimate of the DAG that best explains the observed data. To overcome the challenge that posterior over DAGs is discrete that prohibits gradient optimization, DiBS (Lorch et al., 2021) propose to conduct Stein Variational Gradient Descent (SVGD) (Liu & Wang, 2016b) in the continuous space of a latent probabilistic graph representation.

**Causal (Bayesian) Experimental Design**   Experimental design for causal discovery in a BOED setting was initially explored by Murphy (2001) and Tong & Koller (2001) for discrete variables with single target acquisition. Subsequent research has extended this to continuous variables within the BOED framework (Agrawal et al., 2019; von Kügelgen et al., 2019; Toth et al., 2022; Cho et al., 2016) and alternative frameworks (Kocaoglu et al., 2017a; Gamella & Heinze-Deml, 2020; Ghassami et al., 2018; Olko et al., 2024). Notable approaches in non-BOED settings include those addressing cyclic structures (Mokhtarian et al., 2022) and latent variables (Kocaoglu et al., 2017b). Within the BOED framework, Tigas et al. (2022) proposed a method for selecting single target-state pairs with stochastic batch acquisition, and Tigas et al. (2023) further extended this work to a gradient-based optimization procedure to acquire a set of optimal intervention target-state pairs.  while Sussex et al. (2021) introduced a greedy strategy for selecting multi-target experiments without specifying intervention states. More recently, Annadani et al. (2024b) proposed an adaptive sequential experimental design method for causal structure learning. However, their goal was to minimize the distance between the predicted graph and the ground-truth causal graph, so it doesn't fall under the BOED category. Gao et al. (2024b) proposed a reinforcement learning-based method for sequential experimental design in causal discovery, utilizing Prior Contrastive Estimation (Foster et al., 2021) as a reward function. While innovative, their approach relies on initial observational data and is computationally intensive. In contrast, our method employs direct policy optimization with a differentiable reward function, enabling more efficient training without the need for initial observational data.

## 10 Additional Experiments

### 10.1 Hyperparameters For Policy and Posterior networks

The initial input to the policy network is of shape $(n = n_{int} \times T, d, 2)$, where the last dimension specifies the intervention data (with 0's for interventions that have not been simulated in further stages) and binary intervention masks. The overall process for the policy network includes:

- 1. The input is passed through a fully connected layer, resulting in a shape of $(n_{int} \times T, d, n_{\text{embedding}})$
- 2. The representation is processed through $L$ transformer layers, each consists of:
    - Two multi-head self-attention sublayers, preceded by layer normalization and followed by dropout.
    - Each sublayer applies a fully connected network, preceded by layer normalization and followed by dropout.

Residual connections are applied after each sublayer. This results in an output of shape $(n_{int} \times T, d, n_{embedding})$.

- 3. Max-pooling is applied over the $n_{int} \times T$ dimension, producing a representation of shape $(d, n_{embedding})$.

- 4. The pooled representation is passed through:

  - A target layer, whose output undergoes a Gumbel-softmax transformation with temperature $\tau$, yielding the intervention target vector.
  - A value layer, whose output is scaled in $\min_{val}$ and $\max_{val}$.

The detailed implementation setup is given in Table 2 with the step associated with $\tau$ representing the training steps, and $T$, $n_{step}$ and $n_{envs}$ per training step are the same for the posterior networks:

Table 2: Hyperparameters for the Policy Network

| Hyperparameter | Value |
|---|---|
| Embedding dimension $n_{embedding}$ | 32 |
| Number of transformer layers ($L$) | 4 |
| Key size in self-attention | 16 |
| Number of attention heads | 8 |
| FFN dimensions | $(n_{embedding}, 4 \times n_{embedding}, n_{embedding})$ |
| Activation | ReLU |
| Dropout rate | 0.05 |
| $\max_{val}$ | 10 |
| $\min_{val}$ | -10 |
| $\tau$ | $\min(5 \times 0.9995^{step}, 0.1)$ |
| Initial learning rate | $10^{-4}$ |
| Scheduler | ExponentialLR with $\gamma = 0.8$, step every 1000 training steps |
| $T$ | 10 when $d = 10, 20$ 
 15000 when $d = 30$ |
| $n_{step}$ | 10000 when $d = 10$ 
 15000 when $d = 20, 30$ |
| $n_{env}$ per training step | 10 |

For the posterior networks, the initial input is of shape $(n_{envs}, n_{int} \times T, d, 2)$ with full trajectories being simulated. The process follows the same steps as the policy network up to step 3, resulting in a max-pooled output of shape $(n_{envs}, d, n_{embedding})$.

Starting from step 4, specific to the causal discovery case:

- 4. The pooled representation is passed through:

  - Two independent linear transformations to produce $u$ and $v$, both of shape $(n_{envs}, d, n_{out})$.
  - $u$ and $v$ are normalized using their $l_2$-norm along the last dimension to ensure unit length.

- 5. Compute pairwise logits for all edges:

  - Computes the dot product between every pair of variables $u_i$ and $v_j$, resulting in shape $(n_{envs}, d, d)$.
  - These logits are scaled by a learnable temperature parameter temp, using $logit_{ij} \times exp(\text{temp})$, which is then added with a learnable parameter bias.

The associated implementation setup is given in Table 3:

Table 3: Hyperparameters for the Posterior Network for causal discovery

| Hyperparameter | Value |
|---|---|
| Embedding dimension $n_{\text{embedding}}$ | 128 |
| Number of transformer layers ($L$) | 8 |
| Key size in self-attention | 64 |
| Number of attention heads | 8 |
| FFN dimensions | $(n_{\text{embedding}}, 4 \times n_{\text{embedding}}, n_{\text{embedding}})$ |
| Activation | ReLU |
| Dropout rate | 0.05 |
| bias | -3 |
| temp | 2 |
| Initial LR | $10^{-4}$ |
| Scheduler | ExponentialLR with $\gamma = 0.8$, step every 1000 training steps |

For causal reasoning, starting from step 4:

- 4. The pooled representation is flattened in shape $n_{envs}, d \times n_{\text{embedding}}$, which is fed into the $s(\cdot)$ and $t(\cdot)$ networks as in equation (19), with $n_{\text{trans}}$ transformations in total, ending in shape $n_{envs}, n_{\boldsymbol{z}}$.

The associated implementation setup is given in Table 4:

Table 4: Hyperparameters for the Posterior Network for causal reasoning

| Hyperparameter | Value |
|---|---|
| Embedding dimension $n_{\text{embedding}}$ | 64 |
| Number of transformer layers ($L$) | 8 |
| Key size in self-attention | 16 |
| Number of attention heads | 8 |
| FFN dimensions | $(n_{\text{embedding}}, 4 \times n_{\text{embedding}}, n_{\text{embedding}})$ |
| Activation | ReLU |
| Dropout rate | 0.05 |
| $n_{\text{trans}}$ | 4 |
| $s(\cdot)$ and $t(\cdot)$ dimensions | (256, 256, 256) |
| Initial LR | $1e-4$ |
| Scheduler | ExponentialLR with $\gamma = 0.8$, step every 1000 training steps |

## 10.2 DISTRIBUTION SHIFT IN NOISE

Here we consider the case where the noise distribution is changed from Gaussian to Gumbel, specifically,

$$X_i = f_i(\boldsymbol{X}_{par(i)}, \boldsymbol{\theta}_i) + \epsilon_{X_i}) \tag{24}$$

$$\epsilon_{X_i} \sim N(0, \sigma_i^2) \tag{25}$$

While the policies $\pi$ and the posterior networks $q_{\boldsymbol{\lambda}}(\cdot)$ for previous cases are trained with $\epsilon_{X_i} \sim N(0, 0.3^2)$, we evaluate their performance when applied to data generated with $\epsilon_{X_i} \sim \text{Gumbel}(0, \sigma_i)$ and $\sigma_i^2 \sim \text{InverseGamma}(10, 1)$. To assess the robustness of the trained policies and posterior networks, we also train a separate $q_{\boldsymbol{\lambda}}(\cdot)$ using a random policy on data generated with the shifted noise distribution for causal discovery tasks involving 10 nodes on both Erdős Rényi and Scale Free graphs.

From Table table 5, ACED trained using $T = 10$ outperforms the random policy with the trained posterior across all metrics starting from $T = 8$, demonstrating robustness to the shifted noise distribution.

Table 5: Expected utility and its standard deviation ($\pm$) over 4 different training seeds for policies and posterior networks with $T = 10$. Best results are in bold.

| Number of stages | Metric | ACED (ER) | Random (ER) | ACED (SF) | Random (SF) |
|---|---|---|---|---|---|
| 2 | $\log q$ ($\uparrow$) | $-0.529 \pm 0.034$ | $\mathbf{-0.360 \pm 0.037}$ | $-0.457 \pm 0.017$ | $\mathbf{-0.337 \pm 0.055}$ |
| | $\mathbb{E}$-SHD ($\downarrow$) | $20.64 \pm 0.93$ | $\mathbf{14.99 \pm 1.31}$ | $17.05 \pm 0.51$ | $\mathbf{15.25 \pm 2.87}$ |
| | $F_1$ Score ($\uparrow$) | $0.477 \pm 0.072$ | $\mathbf{0.681 \pm 0.018}$ | $0.343 \pm 0.170$ | $\mathbf{0.549 \pm 0.128}$ |
| 4 | $\log q$ ($\uparrow$) | $-0.328 \pm 0.033$ | $\mathbf{-0.231 \pm 0.015}$ | $-0.296 \pm 0.022$ | $\mathbf{-0.189 \pm 0.047}$ |
| | $\mathbb{E}$-SHD ($\downarrow$) | $14.53 \pm 3.13$ | $\mathbf{11.03 \pm 1.41}$ | $11.65 \pm 1.78$ | $\mathbf{9.20 \pm 2.57}$ |
| | $F_1$ Score ($\uparrow$) | $0.662 \pm 0.077$ | $\mathbf{0.795 \pm 0.016}$ | $0.634 \pm 0.072$ | $\mathbf{0.764 \pm 0.078}$ |
| 6 | $\log q$ ($\uparrow$) | $-0.215 \pm 0.027$ | $\mathbf{-0.188 \pm 0.008}$ | $-0.181 \pm 0.027$ | $\mathbf{-0.133 \pm 0.028}$ |
| | $\mathbb{E}$-SHD ($\downarrow$) | $10.91 \pm 1.88$ | $\mathbf{8.62 \pm 1.33}$ | $7.87 \pm 3.57$ | $\mathbf{6.72 \pm 1.17}$ |
| | $F_1$ Score ($\uparrow$) | $0.778 \pm 0.023$ | $\mathbf{0.822 \pm 0.047}$ | $\mathbf{0.858 \pm 0.056}$ | $0.843 \pm 0.057$ |
| 8 | $\log q$ ($\uparrow$) | $\mathbf{-0.140 \pm 0.013}$ | $-0.167 \pm 0.007$ | $\mathbf{-0.097 \pm 0.022}$ | $-0.116 \pm 0.022$ |
| | $\mathbb{E}$-SHD ($\downarrow$) | $\mathbf{6.60 \pm 0.56}$ | $9.25 \pm 1.41$ | $\mathbf{5.13 \pm 2.46}$ | $6.32 \pm 1.08$ |
| | $F_1$ Score ($\uparrow$) | $\mathbf{0.885 \pm 0.042}$ | $0.829 \pm 0.029$ | $\mathbf{0.957 \pm 0.013}$ | $0.849 \pm 0.060$ |
| 10 | $\log q$ ($\uparrow$) | $\mathbf{-0.110 \pm 0.007}$ | $-0.157 \pm 0.007$ | $\mathbf{-0.066 \pm 0.013}$ | $-0.103 \pm 0.016$ |
| | $\mathbb{E}$-SHD ($\downarrow$) | $\mathbf{5.16 \pm 0.15}$ | $9.04 \pm 1.13$ | $\mathbf{2.52 \pm 0.678}$ | $6.23 \pm 0.49$ |
| | $F_1$ Score ($\uparrow$) | $\mathbf{0.913 \pm 0.030}$ | $0.835 \pm 0.040$ | $\mathbf{0.971 \pm 0.020}$ | $0.868 \pm 0.031$ |

## 10.3 RUNTIME PERFORMANCE

Figure 8 compares the deployment times of various causal Bayesian experimental design methods. These tests were conducted on the same GPU to ensure fair comparison. ACED demonstrates significantly faster inference times, especially as the number of graph nodes increases. This speed is crucial for real-time applications where rapid decision-making is essential.

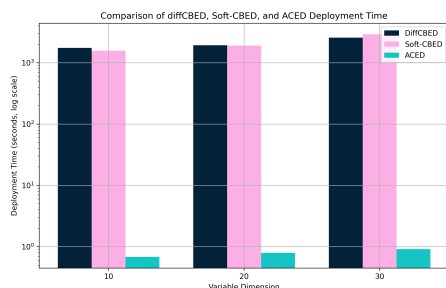

Figure 8: Deployment times for causal Bayesian experimental design methods, evaluated on the same GPU

## 10.4 ADDITIONAL EXPERIMENTAL RESULTS ON SYNTHETIC SCMs

### 10.4.1 RESULTS ON LINEAR SYNTHETIC SCMS

Table 6: Results of different causal experimental design methods across multiple stages on 10-node Erdös Rényi graphs with linear additive noise models. Values show mean $\pm$ standard deviation over 4 random seeds. Best results are in bold.

| Number of stages | Metric | DiffCBED | SoftCBED | Random Policy | ACED (ours) |
|---|---|---|---|---|---|
| 2 | $\log q$ ($\uparrow$) | - | - | $\mathbf{-0.35 \pm 0.02}$ | $-0.52 \pm 0.02$ |
| | $\mathbb{E}$-SHD ($\downarrow$) | $17.30 \pm 0.00$ | $17.22 \pm 0.22$ | $\mathbf{13.50 \pm 4.61}$ | $21.75 \pm 0.83$ |
| | $F_1$ Score ($\uparrow$) | $0.51 \pm 0.00$ | $0.51 \pm 0.01$ | $\mathbf{0.69 \pm 0.08}$ | $0.38 \pm 0.11$ |
| 4 | $\log q$ ($\uparrow$) | - | - | $\mathbf{-0.23 \pm 0.01}$ | $-0.32 \pm 0.02$ |
| | $\mathbb{E}$-SHD ($\downarrow$) | $18.70 \pm 0.28$ | $19.53 \pm 0.56$ | $\mathbf{11.25 \pm 1.92}$ | $15.50 \pm 2.29$ |
| | $F_1$ Score ($\uparrow$) | $0.45 \pm 0.01$ | $0.43 \pm 0.01$ | $\mathbf{0.79 \pm 0.05}$ | $0.68 \pm 0.03$ |
| 6 | $\log q$ ($\uparrow$) | - | - | $-0.20 \pm 0.01$ | $\mathbf{-0.19 \pm 0.02}$ |
| | $\mathbb{E}$-SHD ($\downarrow$) | $18.85 \pm 0.35$ | $19.65 \pm 0.56$ | $\mathbf{10.50 \pm 1.66}$ | $11.00 \pm 1.22$ |
| | $F_1$ Score ($\uparrow$) | $0.44 \pm 0.01$ | $0.42 \pm 0.01$ | $\mathbf{0.83 \pm 0.04}$ | $0.83 \pm 0.03$ |
| 8 | $\log q$ ($\uparrow$) | - | - | $-0.17 \pm 0.01$ | $\mathbf{-0.12 \pm 0.01}$ |
| | $\mathbb{E}$-SHD ($\downarrow$) | $18.00 \pm 0.11$ | $18.15 \pm 0.85$ | $12.00 \pm 1.87$ | $\mathbf{6.50 \pm 2.06}$ |
| | $F_1$ Score ($\uparrow$) | $0.47 \pm 0.01$ | $0.46 \pm 0.03$ | $0.82 \pm 0.03$ | $\mathbf{0.90 \pm 0.03}$ |
| 10 | $\log q$ ($\uparrow$) | - | - | $-0.16 \pm 0.01$ | $\mathbf{-0.08 \pm 0.00}$ |
| | $\mathbb{E}$-SHD ($\downarrow$) | $17.42 \pm 0.33$ | $17.72 \pm 0.77$ | $8.75 \pm 2.28$ | $\mathbf{6.75 \pm 1.09}$ |
| | $F_1$ Score ($\uparrow$) | $0.49 \pm 0.01$ | $0.48 \pm 0.02$ | $0.83 \pm 0.04$ | $\mathbf{0.91 \pm 0.01}$ |

### 10.4.2 RESULTS ON NONLINEAR SYNTHETIC SCMS

Table 7: Results of different causal experimental design methods across multiple stages on 20-node Erdös Rényi graphs with linear additive noise models. Values show mean $\pm$ standard deviation over 4 random seeds. Best results are in bold.

| Number of stages | Metric | DiffCBED | SoftCBED | Random Policy | ACED (ours) |
|---|---|---|---|---|---|
| 2 | $\log q$ ($\uparrow$) | - | - | $-\mathbf{0.27} \pm \mathbf{0.01}$ | $-0.28 \pm 0.02$ |
| | $\mathbb{E}$- SHD ($\downarrow$) | $\mathbf{47.35} \pm \mathbf{0.00}$ | $47.80 \pm 0.46$ | $61.30 \pm 1.13$ | $60.85 \pm 2.72$ |
| | $F_1$ Score ($\uparrow$) | $0.20 \pm 0.00$ | $0.20 \pm 0.01$ | $0.36 \pm 0.06$ | $\mathbf{0.37} \pm \mathbf{0.12}$ |
| 4 | $\log q$ ($\uparrow$) | - | - | $-0.23 \pm 0.01$ | $-\mathbf{0.21} \pm \mathbf{0.01}$ |
| | $\mathbb{E}$- SHD ($\downarrow$) | $\mathbf{48.25} \pm \mathbf{0.00}$ | $48.52 \pm 0.09$ | $55.70 \pm 2.46$ | $51.09 \pm 3.36$ |
| | $F_1$ Score ($\uparrow$) | $0.20 \pm 0.00$ | $0.19 \pm 0.00$ | $0.52 \pm 0.06$ | $\mathbf{0.57} \pm \mathbf{0.04}$ |
| 6 | $\log q$ ($\uparrow$) | - | - | $-0.22 \pm 0.01$ | $-\mathbf{0.20} \pm \mathbf{0.00}$ |
| | $\mathbb{E}$- SHD ($\downarrow$) | $45.70 \pm 0.00$ | $46.75 \pm 0.29$ | $53.93 \pm 3.29$ | $\mathbf{45.50} \pm \mathbf{1.89}$ |
| | $F_1$ Score ($\uparrow$) | $0.24 \pm 0.00$ | $0.21 \pm 0.00$ | $0.58 \pm 0.06$ | $\mathbf{0.65} \pm \mathbf{0.02}$ |
| 8 | $\log q$ ($\uparrow$) | - | - | $-0.22 \pm 0.01$ | $-\mathbf{0.17} \pm \mathbf{0.01}$ |
| | $\mathbb{E}$- SHD ($\downarrow$) | $46.85 \pm 0.00$ | $47.32 \pm 0.12$ | $52.91 \pm 2.94$ | $\mathbf{39.45} \pm \mathbf{1.54}$ |
| | $F_1$ Score ($\uparrow$) | $0.22 \pm 0.00$ | $0.21 \pm 0.01$ | $0.60 \pm 0.07$ | $\mathbf{0.75} \pm \mathbf{0.02}$ |
| 10 | $\log q$ ($\uparrow$) | - | - | $-0.23 \pm 0.02$ | $-\mathbf{0.14} \pm \mathbf{0.00}$ |
| | $\mathbb{E}$- SHD ($\downarrow$) | $47.00 \pm 0.00$ | $47.75 \pm 0.36$ | $56.12 \pm 3.67$ | $\mathbf{34.04} \pm \mathbf{1.11}$ |
| | $F_1$ Score ($\uparrow$) | $0.23 \pm 0.00$ | $0.22 \pm 0.01$ | $0.54 \pm 0.06$ | $\mathbf{0.80} \pm \mathbf{0.01}$ |

Table 8: Results of different causal experimental design methods across multiple stages on 30-node Erdös Rényi graphs with linear additive noise models. Values show mean $\pm$ standard deviation over 4 random seeds. Best results are in bold.

| Number of stages | Metric | DiffCBED | SoftCBED | Random Policy | ACED (ours) |
|---|---|---|---|---|---|
| 2 | $\log q$ ($\uparrow$) | - | - | $-0.29 \pm 0.01$ | $-\mathbf{0.28} \pm \mathbf{0.01}$ |
| | $\mathbb{E}$- SHD ($\downarrow$) | $\mathbf{95.25} \pm \mathbf{7.40}$ | $97.55 \pm 9.40$ | $136.14 \pm 1.34$ | $125.23 \pm 2.63$ |
| | $F_1$ Score ($\uparrow$) | $\mathbf{0.23} \pm \mathbf{0.05}$ | $0.22 \pm 0.04$ | $0.14 \pm 0.02$ | $0.20 \pm 0.03$ |
| 4 | $\log q$ ($\uparrow$) | - | - | $-0.29 \pm 0.01$ | $-\mathbf{0.26} \pm \mathbf{0.01}$ |
| | $\mathbb{E}$- SHD ($\downarrow$) | $98.22 \pm 11.18$ | $\mathbf{94.38} \pm \mathbf{10.33}$ | $132.87 \pm 2.90$ | $124.51 \pm 4.38$ |
| | $F_1$ Score ($\uparrow$) | $0.23 \pm 0.05$ | $\mathbf{0.24} \pm \mathbf{0.03}$ | $0.17 \pm 0.03$ | $0.24 \pm 0.03$ |
| 6 | $\log q$ ($\uparrow$) | - | - | $-0.26 \pm 0.01$ | $-\mathbf{0.25} \pm \mathbf{0.00}$ |
| | $\mathbb{E}$- SHD ($\downarrow$) | $100.17 \pm 13.12$ | $\mathbf{99.75} \pm \mathbf{11.25}$ | $126.32 \pm 1.33$ | $123.34 \pm 3.92$ |
| | $F_1$ Score ($\uparrow$) | $0.23 \pm 0.03$ | $0.23 \pm 0.04$ | $\mathbf{0.29} \pm \mathbf{0.03}$ | $0.29 \pm 0.02$ |
| 8 | $\log q$ ($\uparrow$) | - | - | $-0.25 \pm 0.02$ | $-\mathbf{0.22} \pm \mathbf{0.00}$ |
| | $\mathbb{E}$- SHD ($\downarrow$) | $\mathbf{93.67} \pm \mathbf{10.67}$ | $93.95 \pm 8.15$ | $123.46 \pm 4.87$ | $113.54 \pm 2.38$ |
| | $F_1$ Score ($\uparrow$) | $0.25 \pm 0.03$ | $0.25 \pm 0.05$ | $0.33 \pm 0.05$ | $\mathbf{0.48} \pm \mathbf{0.04}$ |
| 10 | $\log q$ ($\uparrow$) | - | - | $-0.24 \pm 0.01$ | $-\mathbf{0.20} \pm \mathbf{0.01}$ |
| | $\mathbb{E}$- SHD ($\downarrow$) | $\mathbf{92.83} \pm \mathbf{5.88}$ | $96.55 \pm 9.25$ | $123.94 \pm 6.25$ | $106.28 \pm 0.81$ |
| | $F_1$ Score ($\uparrow$) | $0.25 \pm 0.04$ | $0.23 \pm 0.04$ | $0.35 \pm 0.07$ | $\mathbf{0.48} \pm \mathbf{0.01}$ |
| 12 | $\log q$ ($\uparrow$) | - | - | $-0.25 \pm 0.01$ | $-\mathbf{0.19} \pm \mathbf{0.02}$ |
| | $\mathbb{E}$- SHD ($\downarrow$) | $97.72 \pm 10.62$ | $\mathbf{97.55} \pm \mathbf{7.85}$ | $123.60 \pm 6.62$ | $100.53 \pm 9.01$ |
| | $F_1$ Score ($\uparrow$) | $0.25 \pm 0.05$ | $0.25 \pm 0.06$ | $0.34 \pm 0.07$ | $\mathbf{0.57} \pm \mathbf{0.04}$ |
| 14 | $\log q$ ($\uparrow$) | - | - | $-0.23 \pm 0.01$ | $-\mathbf{0.18} \pm \mathbf{0.02}$ |
| | $\mathbb{E}$- SHD ($\downarrow$) | $\mathbf{94.72} \pm \mathbf{11.62}$ | $94.83 \pm 9.17$ | $116.92 \pm 7.86$ | $96.03 \pm 5.36$ |
| | $F_1$ Score ($\uparrow$) | $0.27 \pm 0.03$ | $0.27 \pm 0.05$ | $0.43 \pm 0.08$ | $\mathbf{0.59} \pm \mathbf{0.04}$ |
| 15 | $\log q$ ($\uparrow$) | - | - | $-0.23 \pm 0.02$ | $-\mathbf{0.18} \pm \mathbf{0.00}$ |
| | $\mathbb{E}$- SHD ($\downarrow$) | $97.88 \pm 15.03$ | $97.35 \pm 12.90$ | $117.27 \pm 9.43$ | $\mathbf{93.18} \pm \mathbf{2.19}$ |
| | $F_1$ Score ($\uparrow$) | $0.26 \pm 0.02$ | $0.26 \pm 0.02$ | $0.40 \pm 0.11$ | $\mathbf{0.62} \pm \mathbf{0.02}$ |

Table 9: Results of different causal experimental design methods across multiple stages on 10-node Scale-free graphs with linear additive noise models. Values show mean $\pm$ standard deviation over 4 random seeds. Best results are in bold.

| Number of stages | Metric | DiffCBED | SoftCBED | Random Policy | ACED (ours) |
|---|---|---|---|---|---|
| 2 | $\log q$ ($\uparrow$) | - | - | $-\mathbf{0.36} \pm \mathbf{0.02}$ | $-0.43 \pm 0.03$ |
|  | $\mathbb{E}$- SHD ($\downarrow$) | $\mathbf{10.30} \pm \mathbf{0.00}$ | $10.32 \pm 0.02$ | $13.25 \pm 1.09$ | $14.75 \pm 1.09$ |
|  | $F_1$ Score ($\uparrow$) | $\mathbf{0.58} \pm \mathbf{0.00}$ | $0.58 \pm 0.00$ | $0.54 \pm 0.06$ | $0.39 \pm 0.13$ |
| 4 | $\log q$ ($\uparrow$) | - | - | $-\mathbf{0.21} \pm \mathbf{0.03}$ | $-0.28 \pm 0.02$ |
|  | $\mathbb{E}$- SHD ($\downarrow$) | $11.68 \pm 0.31$ | $11.28 \pm 0.14$ | $\mathbf{9.00} \pm \mathbf{2.00}$ | $11.50 \pm 3.57$ |
|  | $F_1$ Score ($\uparrow$) | $0.49 \pm 0.02$ | $0.51 \pm 0.01$ | $\mathbf{0.76} \pm \mathbf{0.00}$ | $0.72 \pm 0.07$ |
| 6 | $\log q$ ($\uparrow$) | - | - | $-0.16 \pm 0.03$ | $-\mathbf{0.16} \pm \mathbf{0.01}$ |
|  | $\mathbb{E}$- SHD ($\downarrow$) | $10.75 \pm 0.23$ | $11.02 \pm 0.08$ | $11.00 \pm 1.22$ | $\mathbf{7.75} \pm \mathbf{1.48}$ |
|  | $F_1$ Score ($\uparrow$) | $0.55 \pm 0.01$ | $0.54 \pm 0.00$ | $0.77 \pm 0.02$ | $\mathbf{0.79} \pm \mathbf{0.07}$ |
| 8 | $\log q$ ($\uparrow$) | - | - | $-0.14 \pm 0.02$ | $-\mathbf{0.08} \pm \mathbf{0.01}$ |
|  | $\mathbb{E}$- SHD ($\downarrow$) | $10.57 \pm 0.15$ | $11.18 \pm 0.15$ | $8.50 \pm 0.50$ | $\mathbf{5.75} \pm \mathbf{1.64}$ |
|  | $F_1$ Score ($\uparrow$) | $0.57 \pm 0.01$ | $0.53 \pm 0.01$ | $0.75 \pm 0.01$ | $\mathbf{0.85} \pm \mathbf{0.04}$ |
| 10 | $\log q$ ($\uparrow$) | - | - | $-0.12 \pm 0.02$ | $-\mathbf{0.04} \pm \mathbf{0.00}$ |
|  | $\mathbb{E}$- SHD ($\downarrow$) | $10.57 \pm 0.16$ | $11.02 \pm 0.39$ | $8.75 \pm 1.30$ | $\mathbf{5.50} \pm \mathbf{1.12}$ |
|  | $F_1$ Score ($\uparrow$) | $0.57 \pm 0.01$ | $0.54 \pm 0.02$ | $0.76 \pm 0.01$ | $\mathbf{0.85} \pm \mathbf{0.01}$ |

Table 10: Results of different causal experimental design methods across multiple stages on 20-node Scale-free graphs with linear additive noise models. Values show mean $\pm$ standard deviation over 4 random seeds. Best results are in bold.

| Number of stages | Metric | DiffCBED | SoftCBED | Random Policy | ACED (ours) |
|---|---|---|---|---|---|
| 2 | $\log q$ ($\uparrow$) | - | - | $-0.23 \pm 0.03$ | $-\mathbf{0.22} \pm \mathbf{0.02}$ |
|  | $\mathbb{E}$- SHD ($\downarrow$) | $33.05 \pm 0.00$ | $\mathbf{29.80} \pm \mathbf{0.14}$ | $50.67 \pm 1.75$ | $46.68 \pm 1.09$ |
|  | $F_1$ Score ($\uparrow$) | $0.19 \pm 0.00$ | $0.33 \pm 0.00$ | $0.37 \pm 0.08$ | $\mathbf{0.38} \pm \mathbf{0.03}$ |
| 4 | $\log q$ ($\uparrow$) | - | - | $-0.21 \pm 0.01$ | $-\mathbf{0.18} \pm \mathbf{0.01}$ |
|  | $\mathbb{E}$- SHD ($\downarrow$) | $33.20 \pm 0.00$ | $\mathbf{29.65} \pm \mathbf{0.08}$ | $48.07 \pm 2.07$ | $41.35 \pm 2.62$ |
|  | $F_1$ Score ($\uparrow$) | $0.19 \pm 0.00$ | $0.34 \pm 0.00$ | $0.55 \pm 0.03$ | $\mathbf{0.57} \pm \mathbf{0.04}$ |
| 6 | $\log q$ ($\uparrow$) | - | - | $-0.23 \pm 0.02$ | $-\mathbf{0.18} \pm \mathbf{0.01}$ |
|  | $\mathbb{E}$- SHD ($\downarrow$) | $33.20 \pm 0.00$ | $\mathbf{29.82} \pm \mathbf{0.05}$ | $49.83 \pm 2.72$ | $42.02 \pm 2.05$ |
|  | $F_1$ Score ($\uparrow$) | $0.19 \pm 0.00$ | $0.34 \pm 0.00$ | $0.50 \pm 0.03$ | $\mathbf{0.59} \pm \mathbf{0.04}$ |
| 8 | $\log q$ ($\uparrow$) | - | - | $-0.22 \pm 0.03$ | $-\mathbf{0.21} \pm \mathbf{0.01}$ |
|  | $\mathbb{E}$- SHD ($\downarrow$) | $33.20 \pm 0.00$ | $\mathbf{29.60} \pm \mathbf{0.00}$ | $50.41 \pm 3.54$ | $46.31 \pm 0.74$ |
|  | $F_1$ Score ($\uparrow$) | $0.19 \pm 0.00$ | $0.34 \pm 0.00$ | $\mathbf{0.59} \pm \mathbf{0.05}$ | $0.58 \pm 0.02$ |
| 10 | $\log q$ ($\uparrow$) | - | - | $-0.23 \pm 0.01$ | $-\mathbf{0.22} \pm \mathbf{0.02}$ |
|  | $\mathbb{E}$- SHD ($\downarrow$) | $33.15 \pm 0.00$ | $\mathbf{29.73} \pm \mathbf{0.06}$ | $51.68 \pm 0.65$ | $44.94 \pm 1.37$ |
|  | $F_1$ Score ($\uparrow$) | $0.19 \pm 0.00$ | $0.34 \pm 0.00$ | $0.56 \pm 0.01$ | $\mathbf{0.62} \pm \mathbf{0.03}$ |

Table 11: Results of different causal experimental design methods across multiple stages on 30-node Scale-free graphs with linear additive noise models. Values show mean $\pm$ standard deviation over 4 random seeds. Best results are in bold.

| Number of stages | Metric | DiffCBED | SoftCBED | Random Policy | ACED (ours) |
|---|---|---|---|---|---|
| 2 | $\log q$ ($\uparrow$) | - | - | $\mathbf{-0.23 \pm 0.01}$ | $-0.24 \pm 0.01$ |
| | $\mathbb{E}$- SHD ($\downarrow$) | $\mathbf{62.05 \pm 0.00}$ | $62.25 \pm 0.00$ | $121.08 \pm 3.10$ | $89.28 \pm 1.27$ |
| | $F_1$ Score ($\uparrow$) | $\mathbf{0.47 \pm 0.00}$ | $0.46 \pm 0.00$ | $0.29 \pm 0.05$ | $0.26 \pm 0.01$ |
| 4 | $\log q$ ($\uparrow$) | - | - | $-0.22 \pm 0.03$ | $\mathbf{-0.20 \pm 0.01}$ |
| | $\mathbb{E}$- SHD ($\downarrow$) | $\mathbf{64.00 \pm 2.15}$ | $64.80 \pm 0.15$ | $119.19 \pm 9.84$ | $81.28 \pm 2.91$ |
| | $F_1$ Score ($\uparrow$) | $\mathbf{0.46 \pm 0.02}$ | $0.44 \pm 0.00$ | $0.37 \pm 0.02$ | $0.31 \pm 0.01$ |
| 6 | $\log q$ ($\uparrow$) | - | - | $-0.22 \pm 0.02$ | $\mathbf{-0.20 \pm 0.01}$ |
| | $\mathbb{E}$- SHD ($\downarrow$) | $\mathbf{63.05 \pm 0.05}$ | $63.75 \pm 1.95$ | $121.34 \pm 6.19$ | $78.90 \pm 2.52$ |
| | $F_1$ Score ($\uparrow$) | $\mathbf{0.46 \pm 0.00}$ | $0.45 \pm 0.02$ | $0.43 \pm 0.04$ | $0.41 \pm 0.04$ |
| 8 | $\log q$ ($\uparrow$) | - | - | $-0.24 \pm 0.02$ | $\mathbf{-0.18 \pm 0.00}$ |
| | $\mathbb{E}$- SHD ($\downarrow$) | $\mathbf{62.55 \pm 0.15}$ | $63.42 \pm 0.38$ | $128.25 \pm 12.43$ | $78.52 \pm 0.77$ |
| | $F_1$ Score ($\uparrow$) | $\mathbf{0.47 \pm 0.00}$ | $0.46 \pm 0.00$ | $0.41 \pm 0.02$ | $0.46 \pm 0.02$ |
| 10 | $\log q$ ($\uparrow$) | - | - | $-0.24 \pm 0.04$ | $\mathbf{-0.18 \pm 0.00}$ |
| | $\mathbb{E}$- SHD ($\downarrow$) | $\mathbf{62.42 \pm 1.03}$ | $63.92 \pm 0.12$ | $132.44 \pm 18.18$ | $74.70 \pm 1.02$ |
| | $F_1$ Score ($\uparrow$) | $0.47 \pm 0.01$ | $0.45 \pm 0.00$ | $0.45 \pm 0.05$ | $\mathbf{0.50 \pm 0.03}$ |
| 12 | $\log q$ ($\uparrow$) | - | - | $-0.24 \pm 0.02$ | $\mathbf{-0.20 \pm 0.02}$ |
| | $\mathbb{E}$- SHD ($\downarrow$) | $\mathbf{61.62 \pm 0.92}$ | $63.38 \pm 0.23$ | $129.61 \pm 9.26$ | $80.87 \pm 3.38$ |
| | $F_1$ Score ($\uparrow$) | $\mathbf{0.48 \pm 0.01}$ | $0.47 \pm 0.00$ | $0.41 \pm 0.02$ | $0.48 \pm 0.02$ |
| 14 | $\log q$ ($\uparrow$) | - | - | $-0.24 \pm 0.02$ | $\mathbf{-0.16 \pm 0.00}$ |
| | $\mathbb{E}$- SHD ($\downarrow$) | $\mathbf{65.70 \pm 0.60}$ | $66.75 \pm 0.85$ | $137.92 \pm 13.87$ | $76.52 \pm 2.07$ |
| | $F_1$ Score ($\uparrow$) | $0.45 \pm 0.00$ | $0.44 \pm 0.01$ | $0.43 \pm 0.05$ | $\mathbf{0.53 \pm 0.03}$ |
| 15 | $\log q$ ($\uparrow$) | - | - | $-0.24 \pm 0.02$ | $\mathbf{-0.19 \pm 0.00}$ |
| | $\mathbb{E}$- SHD ($\downarrow$) | $\mathbf{63.23 \pm 0.23}$ | $63.70 \pm 1.20$ | $132.24 \pm 12.05$ | $82.09 \pm 1.32$ |
| | $F_1$ Score ($\uparrow$) | $0.47 \pm 0.00$ | $0.46 \pm 0.01$ | $0.46 \pm 0.07$ | $\mathbf{0.52 \pm 0.01}$ |

Table 12: Results of different causal experimental design methods across multiple stages on 10-node Erdös Rényi graphs with nonlinear additive noise models. Values show mean $\pm$ standard deviation over 4 random seeds. Best results are in bold.

| Number of stages | Metric | DiffCBED | SoftCBED | Random Policy | ACED (ours) |
|---|---|---|---|---|---|
| 2 | $\log q$ ($\uparrow$) | - | - | $\mathbf{-0.49 \pm 0.02}$ | $-0.50 \pm 0.00$ |
| | $\mathbb{E}$- SHD ($\downarrow$) | $23.44 \pm 0.40$ | $26.45 \pm 0.00$ | $\mathbf{18.53 \pm 0.32}$ | $18.72 \pm 0.40$ |
| | $F_1$ Score ($\uparrow$) | $0.42 \pm 0.00$ | $0.35 \pm 0.00$ | $\mathbf{0.43 \pm 0.06}$ | $0.37 \pm 0.06$ |
| 4 | $\log q$ ($\uparrow$) | - | - | $\mathbf{-0.36 \pm 0.04}$ | $-0.41 \pm 0.04$ |
| | $\mathbb{E}$- SHD ($\downarrow$) | $22.04 \pm 0.13$ | $22.67 \pm 0.00$ | $\mathbf{19.22 \pm 1.21}$ | $20.19 \pm 0.41$ |
| | $F_1$ Score ($\uparrow$) | $\mathbf{0.49 \pm 0.01}$ | $0.46 \pm 0.00$ | $0.45 \pm 0.06$ | $0.39 \pm 0.03$ |
| 6 | $\log q$ ($\uparrow$) | - | - | $\mathbf{-0.37 \pm 0.07}$ | $-0.40 \pm 0.04$ |
| | $\mathbb{E}$- SHD ($\downarrow$) | $22.20 \pm 0.19$ | $21.94 \pm 0.00$ | $\mathbf{18.27 \pm 0.93}$ | $18.37 \pm 0.79$ |
| | $F_1$ Score ($\uparrow$) | $\mathbf{0.49 \pm 0.00}$ | $0.47 \pm 0.00$ | $0.47 \pm 0.16$ | $0.42 \pm 0.11$ |
| 8 | $\log q$ ($\uparrow$) | - | - | $\mathbf{-0.32 \pm 0.04}$ | $-0.35 \pm 0.00$ |
| | $\mathbb{E}$- SHD ($\downarrow$) | $22.09 \pm 0.04$ | $22.08 \pm 0.00$ | $\mathbf{17.37 \pm 1.32}$ | $18.38 \pm 0.29$ |
| | $F_1$ Score ($\uparrow$) | $0.50 \pm 0.00$ | $0.48 \pm 0.00$ | $\mathbf{0.54 \pm 0.06}$ | $0.49 \pm 0.02$ |
| 10 | $\log q$ ($\uparrow$) | - | - | $\mathbf{-0.35 \pm 0.06}$ | $-0.38 \pm 0.02$ |
| | $\mathbb{E}$- SHD ($\downarrow$) | $21.86 \pm 0.34$ | $22.36 \pm 0.00$ | $\mathbf{17.39 \pm 0.93}$ | $17.92 \pm 0.46$ |
| | $F_1$ Score ($\uparrow$) | $0.50 \pm 0.00$ | $0.46 \pm 0.00$ | $\mathbf{0.50 \pm 0.10}$ | $0.44 \pm 0.01$ |

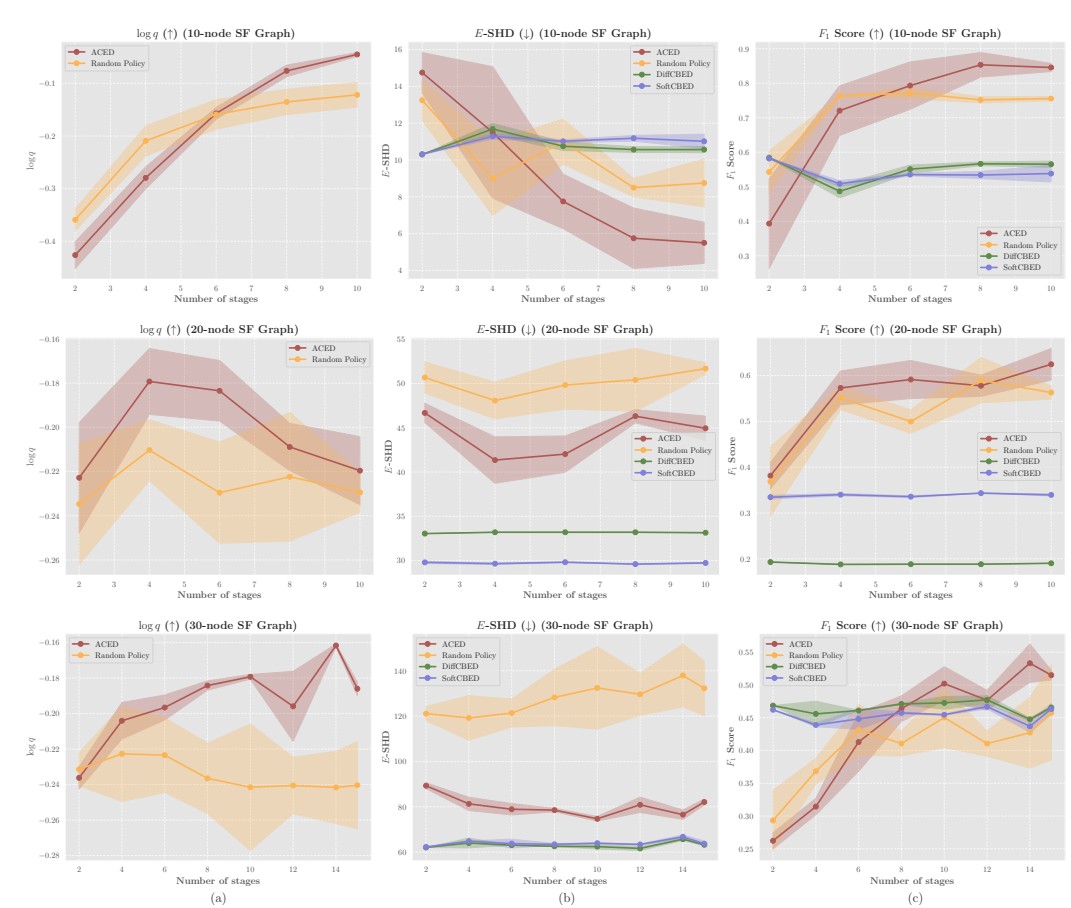

Figure 9: Results on linear SCM with SF graphs

Table 13: Results of different causal experimental design methods across multiple stages on 20-node Erdös Rényi graphs with nonlinear additive noise models. Values show mean ± standard deviation over 4 random seeds. Best results are in bold.

| Number of stages | Metric | DiffCBED | SoftCBED | Random Policy | ACED (ours) |
|---|---|---|---|---|---|
| 2 | $\log q$ ($\uparrow$) | - | - | $\mathbf{0.26 \pm 0.00}$ | $0.26 \pm 0.01$ |
| | $\mathbb{E}$- SHD ($\downarrow$) | $64.33 \pm 0.00$ | $80.50 \pm 0.00$ | $46.80 \pm 0.98$ | $\mathbf{45.54 \pm 1.13}$ |
| | $F_1$ Score ($\uparrow$) | $0.30 \pm 0.00$ | $0.33 \pm 0.00$ | $0.36 \pm 0.04$ | $\mathbf{0.37 \pm 0.05}$ |
| 4 | $\log q$ ($\uparrow$) | - | - | $0.22 \pm 0.01$ | $\mathbf{0.25 \pm 0.00}$ |
| | $\mathbb{E}$- SHD ($\downarrow$) | $64.33 \pm 0.00$ | $101.00 \pm 0.00$ | $48.30 \pm 2.23$ | $\mathbf{48.09 \pm 0.25}$ |
| | $F_1$ Score ($\uparrow$) | $0.30 \pm 0.00$ | $0.29 \pm 0.00$ | $\mathbf{0.40 \pm 0.02}$ | $0.38 \pm 0.02$ |
| 6 | $\log q$ ($\uparrow$) | - | - | $0.26 \pm 0.00$ | $\mathbf{0.26 \pm 0.00}$ |
| | $\mathbb{E}$- SHD ($\downarrow$) | $64.33 \pm 0.00$ | $101.00 \pm 0.00$ | $\mathbf{49.25 \pm 1.42}$ | $51.79 \pm 0.21$ |
| | $F_1$ Score ($\uparrow$) | $0.30 \pm 0.00$ | $0.29 \pm 0.00$ | $\mathbf{0.34 \pm 0.01}$ | $0.33 \pm 0.02$ |
| 8 | $\log q$ ($\uparrow$) | - | - | $0.23 \pm 0.00$ | $\mathbf{0.25 \pm 0.01}$ |
| | $\mathbb{E}$- SHD ($\downarrow$) | $64.33 \pm 0.00$ | $101.00 \pm 0.00$ | $\mathbf{43.69 \pm 0.87}$ | $43.84 \pm 0.77$ |
| | $F_1$ Score ($\uparrow$) | $0.30 \pm 0.00$ | $0.29 \pm 0.00$ | $0.40 \pm 0.02$ | $\mathbf{0.47 \pm 0.00}$ |
| 10 | $\log q$ ($\uparrow$) | - | - | $0.24 \pm 0.01$ | $\mathbf{0.25 \pm 0.00}$ |
| | $\mathbb{E}$- SHD ($\downarrow$) | $64.33 \pm 0.00$ | $101.00 \pm 0.00$ | $\mathbf{48.49 \pm 2.82}$ | $51.62 \pm 1.21$ |
| | $F_1$ Score ($\uparrow$) | $0.30 \pm 0.00$ | $0.29 \pm 0.00$ | $0.37 \pm 0.04$ | $\mathbf{0.38 \pm 0.01}$ |

Table 14: Results of different causal experimental design methods across multiple stages on 30-node Erdös Rényi graphs with nonlinear additive noise models. Values show mean $\pm$ standard deviation over 4 random seeds. Best results are in bold.

| Number of stages | Metric | DiffCBED | SoftCBED | Random Policy | ACED (ours) |
|---|---|---|---|---|---|
| 2 | $\log q$ ($\uparrow$) | - | - | $-0.26 \pm 0.00$ | $-0.27 \pm 0.00$ |
| | $\mathbb{E}$-SHD ($\downarrow$) | $152.59 \pm 3.90$ | $152.95 \pm 4.84$ | $\mathbf{132.53 \pm 0.90}$ | $134.63 \pm 0.20$ |
| | $F_1$ Score ($\uparrow$) | $\mathbf{0.23 \pm 0.01}$ | $0.23 \pm 0.01$ | $0.11 \pm 0.01$ | $0.15 \pm 0.02$ |
| 4 | $\log q$ ($\uparrow$) | - | - | $-0.25 \pm 0.00$ | $\mathbf{-0.25 \pm 0.00}$ |
| | $\mathbb{E}$-SHD ($\downarrow$) | $149.37 \pm 6.40$ | $146.56 \pm 2.66$ | $133.28 \pm 0.88$ | $\mathbf{130.04 \pm 0.17}$ |
| | $F_1$ Score ($\uparrow$) | $\mathbf{0.23 \pm 0.01}$ | $0.22 \pm 0.01$ | $0.14 \pm 0.01$ | $0.21 \pm 0.02$ |
| 6 | $\log q$ ($\uparrow$) | - | - | $-0.25 \pm 0.00$ | $-0.25 \pm 0.00$ |
| | $\mathbb{E}$-SHD ($\downarrow$) | $150.19 \pm 2.31$ | $144.09 \pm 2.58$ | $\mathbf{126.90 \pm 0.17}$ | $127.48 \pm 0.64$ |
| | $F_1$ Score ($\uparrow$) | $\mathbf{0.23 \pm 0.00}$ | $0.22 \pm 0.01$ | $0.17 \pm 0.03$ | $0.15 \pm 0.01$ |
| 8 | $\log q$ ($\uparrow$) | - | - | $-0.24 \pm 0.00$ | $-0.26 \pm 0.00$ |
| | $\mathbb{E}$-SHD ($\downarrow$) | $149.76 \pm 4.92$ | $142.99 \pm 3.37$ | $130.17 \pm 0.52$ | $\mathbf{129.73 \pm 0.40}$ |
| | $F_1$ Score ($\uparrow$) | $\mathbf{0.22 \pm 0.00}$ | $0.22 \pm 0.01$ | $0.18 \pm 0.00$ | $0.18 \pm 0.00$ |
| 10 | $\log q$ ($\uparrow$) | - | - | $-0.24 \pm 0.00$ | $-0.24 \pm 0.00$ |
| | $\mathbb{E}$-SHD ($\downarrow$) | $145.67 \pm 5.26$ | $140.95 \pm 4.49$ | $\mathbf{129.93 \pm 0.60}$ | $130.14 \pm 0.22$ |
| | $F_1$ Score ($\uparrow$) | $0.21 \pm 0.01$ | $\mathbf{0.22 \pm 0.01}$ | $0.19 \pm 0.02$ | $0.21 \pm 0.01$ |
| 12 | $\log q$ ($\uparrow$) | - | - | $-0.25 \pm 0.00$ | $-0.26 \pm 0.00$ |
| | $\mathbb{E}$-SHD ($\downarrow$) | $143.59 \pm 6.04$ | $141.47 \pm 4.04$ | $\mathbf{133.20 \pm 0.79}$ | $136.35 \pm 0.32$ |
| | $F_1$ Score ($\uparrow$) | $0.20 \pm 0.01$ | $\mathbf{0.21 \pm 0.01}$ | $0.13 \pm 0.02$ | $0.16 \pm 0.03$ |
| 14 | $\log q$ ($\uparrow$) | - | - | $-0.25 \pm 0.00$ | $-0.25 \pm 0.00$ |
| | $\mathbb{E}$-SHD ($\downarrow$) | $142.15 \pm 6.31$ | $140.65 \pm 4.15$ | $132.30 \pm 0.99$ | $\mathbf{129.30 \pm 0.54}$ |
| | $F_1$ Score ($\uparrow$) | $0.20 \pm 0.01$ | $\mathbf{0.20 \pm 0.01}$ | $0.18 \pm 0.02$ | $0.17 \pm 0.00$ |
| 15 | $\log q$ ($\uparrow$) | - | - | $-0.25 \pm 0.00$ | $-0.26 \pm 0.00$ |
| | $\mathbb{E}$-SHD ($\downarrow$) | $137.67 \pm 1.74$ | $138.85 \pm 4.12$ | $\mathbf{132.05 \pm 0.53}$ | $135.92 \pm 0.20$ |
| | $F_1$ Score ($\uparrow$) | $0.18 \pm 0.02$ | $\mathbf{0.21 \pm 0.01}$ | $0.17 \pm 0.02$ | $0.18 \pm 0.01$ |

Table 15: Results of different causal experimental design methods across multiple stages on 10-node Scale-free graphs with nonlinear additive noise models. Values show mean $\pm$ standard deviation over 4 random seeds. Best results are in bold.

| Number of stages | Metric | DiffCBED | SoftCBED | Random Policy | ACED (ours) |
|---|---|---|---|---|---|
| 2 | $\log q$ ($\uparrow$) | - | - | $-0.14 \pm 0.01$ | $-0.42 \pm 0.04$ |
| | $\mathbb{E}$-SHD ($\downarrow$) | $9.41 \pm 0.35$ | $9.56 \pm 0.06$ | $\mathbf{7.35 \pm 0.51}$ | $15.54 \pm 0.77$ |
| | $F_1$ Score ($\uparrow$) | $0.66 \pm 0.01$ | $0.67 \pm 0.00$ | $\mathbf{0.85 \pm 0.01}$ | $0.56 \pm 0.06$ |
| 4 | $\log q$ ($\uparrow$) | - | - | $-0.16 \pm 0.01$ | $-0.28 \pm 0.03$ |
| | $\mathbb{E}$-SHD ($\downarrow$) | $10.15 \pm 0.47$ | $9.07 \pm 0.06$ | $\mathbf{8.67 \pm 0.60}$ | $11.01 \pm 0.77$ |
| | $F_1$ Score ($\uparrow$) | $0.65 \pm 0.01$ | $0.70 \pm 0.01$ | $\mathbf{0.83 \pm 0.01}$ | $0.70 \pm 0.02$ |
| 6 | $\log q$ ($\uparrow$) | - | - | $-0.11 \pm 0.01$ | $-0.18 \pm 0.01$ |
| | $\mathbb{E}$-SHD ($\downarrow$) | $10.02 \pm 0.33$ | $9.28 \pm 0.18$ | $\mathbf{6.39 \pm 0.32}$ | $7.39 \pm 0.33$ |
| | $F_1$ Score ($\uparrow$) | $0.67 \pm 0.01$ | $0.70 \pm 0.02$ | $\mathbf{0.87 \pm 0.02}$ | $0.83 \pm 0.01$ |
| 8 | $\log q$ ($\uparrow$) | - | - | $-0.13 \pm 0.01$ | $\mathbf{-0.10 \pm 0.01}$ |
| | $\mathbb{E}$-SHD ($\downarrow$) | $10.17 \pm 0.57$ | $9.92 \pm 0.71$ | $6.66 \pm 0.34$ | $\mathbf{5.18 \pm 0.16}$ |
| | $F_1$ Score ($\uparrow$) | $0.67 \pm 0.02$ | $0.69 \pm 0.04$ | $0.84 \pm 0.03$ | $\mathbf{0.90 \pm 0.01}$ |
| 10 | $\log q$ ($\uparrow$) | - | - | $-0.11 \pm 0.01$ | $-0.12 \pm 0.01$ |
| | $\mathbb{E}$-SHD ($\downarrow$) | $11.58 \pm 1.53$ | $10.79 \pm 0.43$ | $6.07 \pm 0.83$ | $\mathbf{5.86 \pm 0.29}$ |
| | $F_1$ Score ($\uparrow$) | $0.65 \pm 0.03$ | $0.67 \pm 0.01$ | $\mathbf{0.89 \pm 0.02}$ | $0.86 \pm 0.01$ |

Table 16: Results of different causal experimental design methods across multiple stages on 20-node Scale-free graphs with nonlinear additive noise models. Values show mean $\pm$ standard deviation over 4 random seeds. Best results are in bold.

| Number of stages | Metric | DiffCBED | SoftCBED | Random Policy | ACED (ours) |
|---|---|---|---|---|---|
| 2 | $\log q$ ($\uparrow$) | - | - | $\mathbf{-0.09 \pm 0.00}$ | $-0.11 \pm 0.00$ |
| | $\mathbb{E}$- SHD ($\downarrow$) | $34.50 \pm 0.69$ | $39.43 \pm 0.00$ | $\mathbf{23.95 \pm 1.02}$ | $25.80 \pm 0.24$ |
| | $F_1$ Score ($\uparrow$) | $0.56 \pm 0.01$ | $0.53 \pm 0.00$ | $\mathbf{0.77 \pm 0.02}$ | $0.74 \pm 0.01$ |
| 4 | $\log q$ ($\uparrow$) | - | - | $-0.11 \pm 0.01$ | $\mathbf{-0.10 \pm 0.00}$ |
| | $\mathbb{E}$- SHD ($\downarrow$) | $37.57 \pm 1.15$ | $42.96 \pm 0.00$ | $24.43 \pm 0.64$ | $\mathbf{23.97 \pm 0.20}$ |
| | $F_1$ Score ($\uparrow$) | $0.52 \pm 0.01$ | $0.49 \pm 0.00$ | $0.73 \pm 0.01$ | $\mathbf{0.75 \pm 0.00}$ |
| 6 | $\log q$ ($\uparrow$) | - | - | $\mathbf{-0.08 \pm 0.00}$ | $-0.09 \pm 0.00$ |
| | $\mathbb{E}$- SHD ($\downarrow$) | $38.97 \pm 1.32$ | $45.37 \pm 0.00$ | $21.28 \pm 0.41$ | $\mathbf{20.77 \pm 0.11}$ |
| | $F_1$ Score ($\uparrow$) | $0.51 \pm 0.02$ | $0.49 \pm 0.00$ | $0.79 \pm 0.01$ | $\mathbf{0.79 \pm 0.00}$ |
| 8 | $\log q$ ($\uparrow$) | - | - | $-0.08 \pm 0.01$ | $\mathbf{-0.08 \pm 0.00}$ |
| | $\mathbb{E}$- SHD ($\downarrow$) | $40.55 \pm 1.02$ | $47.57 \pm 0.00$ | $20.86 \pm 1.50$ | $\mathbf{19.84 \pm 0.37}$ |
| | $F_1$ Score ($\uparrow$) | $0.50 \pm 0.02$ | $0.49 \pm 0.00$ | $0.80 \pm 0.01$ | $\mathbf{0.81 \pm 0.01}$ |
| 10 | $\log q$ ($\uparrow$) | - | - | $-0.09 \pm 0.01$ | $\mathbf{-0.08 \pm 0.00}$ |
| | $\mathbb{E}$- SHD ($\downarrow$) | $40.75 \pm 0.43$ | $45.23 \pm 0.00$ | $22.57 \pm 2.34$ | $\mathbf{20.72 \pm 0.31}$ |
| | $F_1$ Score ($\uparrow$) | $0.50 \pm 0.01$ | $0.51 \pm 0.00$ | $0.80 \pm 0.02$ | $\mathbf{0.82 \pm 0.01}$ |

Table 17: Results of different causal experimental design methods across multiple stages on 30-node Scale-free graphs with nonlinear additive noise models. Values show mean $\pm$ standard deviation over 4 random seeds. Best results are in bold.

| Number of stages | Metric | DiffCBED | SoftCBED | Random Policy | ACED (ours) |
|---|---|---|---|---|---|
| 2 | $\log q$ ($\uparrow$) | - | - | $\mathbf{-0.14 \pm 0.00}$ | $-0.17 \pm 0.00$ |
| | $\mathbb{E}$- SHD ($\downarrow$) | $103.98 \pm 1.78$ | $99.22 \pm 0.00$ | $\mathbf{75.57 \pm 1.49}$ | $82.80 \pm 0.90$ |
| | $F_1$ Score ($\uparrow$) | $0.45 \pm 0.01$ | $0.45 \pm 0.00$ | $\mathbf{0.56 \pm 0.02}$ | $0.53 \pm 0.02$ |
| 4 | $\log q$ ($\uparrow$) | - | - | $-0.14 \pm 0.00$ | $\mathbf{-0.14 \pm 0.00}$ |
| | $\mathbb{E}$- SHD ($\downarrow$) | $105.67 \pm 1.75$ | $101.96 \pm 0.00$ | $\mathbf{78.68 \pm 2.34}$ | $80.77 \pm 1.50$ |
| | $F_1$ Score ($\uparrow$) | $0.39 \pm 0.01$ | $0.41 \pm 0.00$ | $0.50 \pm 0.04$ | $\mathbf{0.61 \pm 0.01}$ |
| 6 | $\log q$ ($\uparrow$) | - | - | $-0.13 \pm 0.00$ | $\mathbf{-0.13 \pm 0.00}$ |
| | $\mathbb{E}$- SHD ($\downarrow$) | $100.52 \pm 1.52$ | $101.29 \pm 0.00$ | $77.52 \pm 0.75$ | $\mathbf{73.02 \pm 0.91}$ |
| | $F_1$ Score ($\uparrow$) | $0.35 \pm 0.00$ | $0.38 \pm 0.00$ | $0.58 \pm 0.03$ | $\mathbf{0.64 \pm 0.01}$ |
| 8 | $\log q$ ($\uparrow$) | - | - | $-0.14 \pm 0.01$ | $\mathbf{-0.12 \pm 0.00}$ |
| | $\mathbb{E}$- SHD ($\downarrow$) | $98.61 \pm 0.81$ | $101.46 \pm 0.00$ | $78.87 \pm 3.12$ | $\mathbf{66.99 \pm 0.50}$ |
| | $F_1$ Score ($\uparrow$) | $0.34 \pm 0.00$ | $0.37 \pm 0.00$ | $0.59 \pm 0.03$ | $\mathbf{0.67 \pm 0.00}$ |
| 10 | $\log q$ ($\uparrow$) | - | - | $-0.12 \pm 0.00$ | $\mathbf{-0.11 \pm 0.00}$ |
| | $\mathbb{E}$- SHD ($\downarrow$) | $98.06 \pm 1.20$ | $102.81 \pm 0.00$ | $69.34 \pm 0.74$ | $\mathbf{67.81 \pm 0.31}$ |
| | $F_1$ Score ($\uparrow$) | $0.33 \pm 0.00$ | $0.37 \pm 0.00$ | $0.68 \pm 0.01$ | $\mathbf{0.71 \pm 0.01}$ |
| 12 | $\log q$ ($\uparrow$) | - | - | $-0.13 \pm 0.00$ | $\mathbf{-0.11 \pm 0.00}$ |
| | $\mathbb{E}$- SHD ($\downarrow$) | $100.44 \pm 2.73$ | $104.92 \pm 0.00$ | $76.00 \pm 1.41$ | $\mathbf{63.83 \pm 0.10}$ |
| | $F_1$ Score ($\uparrow$) | $0.35 \pm 0.00$ | $0.36 \pm 0.00$ | $0.66 \pm 0.01$ | $\mathbf{0.73 \pm 0.00}$ |
| 14 | $\log q$ ($\uparrow$) | - | - | $-0.12 \pm 0.01$ | $\mathbf{-0.10 \pm 0.00}$ |
| | $\mathbb{E}$- SHD ($\downarrow$) | $101.68 \pm 2.82$ | $106.29 \pm 0.00$ | $75.97 \pm 3.66$ | $\mathbf{62.69 \pm 0.70}$ |
| | $F_1$ Score ($\uparrow$) | $0.34 \pm 0.01$ | $0.35 \pm 0.00$ | $0.65 \pm 0.01$ | $\mathbf{0.74 \pm 0.00}$ |
| 15 | $\log q$ ($\uparrow$) | - | - | $-0.12 \pm 0.00$ | $\mathbf{-0.11 \pm 0.00}$ |
| | $\mathbb{E}$- SHD ($\downarrow$) | $102.97 \pm 3.13$ | $108.97 \pm 0.00$ | $77.39 \pm 2.41$ | $\mathbf{67.53 \pm 0.27}$ |
| | $F_1$ Score ($\uparrow$) | $0.33 \pm 0.01$ | $0.34 \pm 0.00$ | $0.69 \pm 0.00$ | $\mathbf{0.75 \pm 0.01}$ |

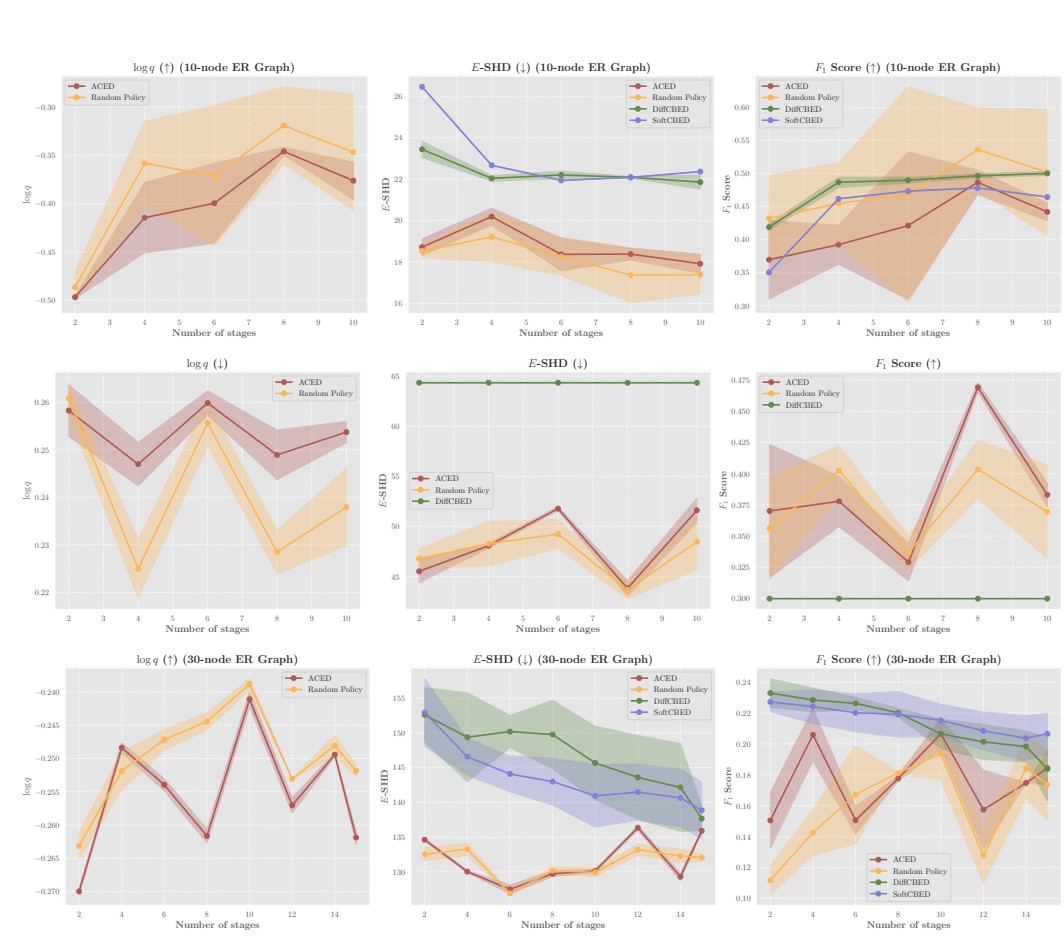

Figure 10: Results on nonlinear SCM with ER graphs

