# OpenReview forum: "Adaptive Causal Experimental Design: Amortizing Sequential Bayesian Experimental Design for Causal Models"
_ICLR.cc/2025/Conference — Submitted to ICLR 2025_

### Official Review · Reviewer_CYhw · 2024-10-24

**Soundness:** 3
**Presentation:** 2
**Contribution:** 2
**Rating:** 5
**Confidence:** 4

**Summary:**

The paper explores methods to enhance causal discovery and reasoning, especially when dealing with costly interventional data. The authors introduce Adaptive Causal Experimental Design (ACED), a new framework that aims to produce non-myopic (long-term beneficial) interventions. The goal is to maximize the Expected Information Gain (EIG) throughout a sequence of experiments rather than focusing solely on the immediate next experiment. ACED uses a policy network (leveraging a transformer-based architecture) to predict the most informative interventions. This policy is trained offline, allowing quick decision-making during actual deployment. The authors also texted ACED on both simulated datasets and real datasets.

**Strengths:**

1. The design of policy network is interesting
2. Some theoretical analysis is provided.

**Weaknesses:**

1. The notations are not consistent, which makes it hard to read. For example, in equation (3), (4), the notation of $h_t$ and $\xi$ are not consistent to definition in line147.
2. As far as I am concerned, the proposed model does not has constraint on DAG, which means it may result in cycles.
3. The authors show that the proposed model is optimizing EIG at each step with variational lower bound, however, there is no analysis on non-asymptotic convergence to the ground truth given enough interventional samples.
4. As the authors mentioned in the discussion section, the model is proposed to relive real world budgets such as time and cost. Some metric to demonstrate this could be more convincing. For example, compare the time of GRU experiments with ACED's policy network.
5. The authors compared ACED with 3 baselines including a random policy. I wonder if the authors could include some baseline that does not fall into the BOED category such as [1] which is also discussed in the related works section in the appendix.

[1] Annadani, Yashas, Panagiotis Tigas, Stefan Bauer, and Adam Foster. "Amortized Active Causal Induction with Deep Reinforcement Learning." arXiv preprint arXiv:2405.16718 (2024).

**Questions:**

See weakness for major questions.
1. Is $Z$ independent to $\mathcal{D}$ given $G, \theta$?
2. Is $\theta$ independent to $G$?
3. I wonder if there is any justifications on the choose of the neural network architectures. For example, the ablation study on the hyper parameters of the transformers could be attached or any suggestion on the selection of parameters. Also, why normalizing flow is preferred?
4. The authors mentioned there could be bias in the EIG estimation in the discussion section. Could the authors elaborate more on how much the bias could aff5ect the model's performance?
5. The actual computation of posterior is replaced by $\pi$ in ACED. I wonder if the author can show the advantage of $\pi$ over actual computation of posteriors with an ablation study on simple simulated graphs?

---

> ### Author Response · Authors · 2024-11-24
> **Response to Reviewer CYhw (Part 1/2)**
>
> We thank the reviewer for the constructive and helpful feedback. We address each of your concerns and questions below:
>
> 1. __Inconsistent notations__:
>     Thank you for pointing this out, we have revised Equations (3)–(6) to ensure consistency.
>
> 2. __DAG constraint__:
>     Currently, we do not enforce DAG constraints. However, we have tested the DAG constraint approach proposed in [1]. While it slightly reduces the number of non-DAGs, the effect is marginal and comes at the cost of increased computational overhead and higher loss. Our experiments suggest that reducing the loss in $\log q$ is sufficient to efficiently generate DAGs, provided that the training samples and steps are adequate.
>
> 3. __No asymptotic convergence analysis__:
>     Thank you for raising this concern. In the Bayesian causal discovery setting, the goal is to obtain a posterior distribution centered on the true causal graph given a sufficiently large number of interventional samples, with minimal uncertainty. In our manuscript, we theoretically demonstrate that our proposed method is non-myopic, effectively reducing the largest uncertainty over a sequence of interventional experiments compared to existing active learning methods. However, we acknowledge the importance of asymptotic convergence analysis and leave this aspect for future work.
>
> 4. __Real-time deployment of our proposed method__:
> Figure 8 in the revised manuscript compares the deployment times of our proposed method, ACED, with two baselines, DiffCBED and SoftCBED, across three variable dimensions (10, 20, and 30) on linear synthetic SCMs with Erdős–Rényi graphs. The times are presented on a logarithmic scale to highlight the substantial differences in magnitudes. The bar plot demonstrates that ACED achieves significantly faster deployment times compared to the baseline methods.
>
> 5. __Comparison with CAASL__:
> Thank you for pointing this out; we would like to clarify. We did not include a comparison with CAASL in our work because the code was not publicly available at the time of submission. Although the authors have since kindly shared the code, additional effort is required to ensure a fair and meaningful comparison. Below, we highlight the key differences and outline our plans for future comparisons:
>
> * __Scope of the approach__: Our method provides a unified framework for designing optimal policies to reduce uncertainty for general causal queries, whereas CAASL focuses specifically on causal discovery.
>
> * __Reward computation__:
> CAASL computes rewards by sampling from the variational posterior $q$ and comparing the samples to the data-generating graph $G$. In contrast, our method evaluates the density of $G$, which is potentially more memory- and computationally-efficient.
>
> * __Training approach__:
> CAASL uses Soft Actor-Critic (SAC) to train the policy, which increases training time but does not require a differentiable reward function. Our approach, on the other hand, relies on a differentiable likelihood for policy optimization.
>
>     Additionally, there are other setup differences, such as the inclusion of initial observations $\mathcal{D}$, the data-generating process, the posterior inference algorithm, and whether the pretrained AVICI model is fine-tuned. In future work, we will ensure a fair and comprehensive comparison with CAASL by carefully aligning these experimental setups.
>
> 6. __Is $\boldsymbol{z}$ independent of $\mathcal{D}$ given $G, \boldsymbol{\theta}$__:
>     Yes, thank you for pointing this out. We have updated Equations (5) and (6) in the manuscript to clarify this. Specifically, $\boldsymbol{z}$ represents a prediction quantity of interest derived from $G$, $\boldsymbol{\theta}$, and $\boldsymbol{\epsilon\_z}$. For example, in Figure 5, $\boldsymbol{z} = {\boldsymbol{x}\_3, \boldsymbol{x}\_4 \mid \text{do}(\boldsymbol{x}\_2 = 2)}$, where:
>     $$\boldsymbol{x}\_3 = \boldsymbol{x}\_1 \times \theta\_{13} + 2 \times \theta\_{23} + \epsilon\_3, \quad \boldsymbol{x}\_4 = 2 \times \theta\_{24} + \boldsymbol{x}\_3 \times \theta\_{34} + \epsilon\_4 .$$
>     Thus, the belief about $\boldsymbol{z}$ is updated through beliefs about $G$ and $\boldsymbol{\theta}$, as reflected in Equations (5) and (6).
>     Here, $\mathcal{D}$ denotes observational data obtained prior to performing interventions, which informs belief updates on $G$ and $\boldsymbol{\theta}$. Without $\mathcal{D}$, the relationship would be:
> $$
>     p(\boldsymbol{z} ) = \sum_G \int_{\boldsymbol{\theta}} p(\boldsymbol{z}| G, \boldsymbol{\theta}) p(G) p(\boldsymbol{\theta} |G )\, \text{d}\boldsymbol{\theta}.
> $$

---

> > ### Author Response · Authors · 2024-11-24
> > **Response to Reviewer CYhw (Part 2/2)**
> >
> > 7. __Is $\boldsymbol{\theta}$ independent to $G$__:
> >     To clarify, $\boldsymbol{\theta}$ is not independent of $G$, as $\boldsymbol{\theta}$ represents the edge parameters, with edge connectivity determined by $G$. For example, in Figure 5, under the given $G$, there is no connection from node 1 to node 4, which implies $\theta_{14} = 0$. In more general cases, where there is uncertainty about edge connectivity, $p(\boldsymbol{\theta})$ differs from $p(\boldsymbol{\theta} \mid G)$. We have updated Section 3.1 in the revised manuscript to clarify this point.
> >
> > 8. __Justification on neural network structures and using Normalizing Flows__:
> >     For causal discovery, where $\boldsymbol{z} = G$, the neural network used for the variational posterior approximator $q_{\boldsymbol{\lambda}}(G \mid \mathcal{D}, f_{\boldsymbol{\phi}}(\boldsymbol{h}_T))$ follows the setup described in [2]. For causal reasoning, where $\boldsymbol{z}$ is continuous, the neural network structure is based on [3].
> >     Normalizing Flows (NFs) are employed because Equation (9) requires maximization over $\log q$, and NFs enable exact density evaluation. That said, alternative generative models, such as Gaussian Mixture Models (GMMs) and autoregressive models, which also support exact density estimation, could be considered for future investigation in this context.
> >
> > 9. __Bias in the EIG estimation in the discussion section__:
> >     Our approach trains a policy to maximize the EIG lower bound, which serves as an estimator of the exact EIG. This lower bound is tight only if we achieve a perfect posterior approximation for all $G$, $\boldsymbol{\theta}$, and $\boldsymbol{y}$ pairs. A tighter lower bound can be obtained by optimizing Equation (9). However, since we use independent Bernoulli distributions for $G$ and normalizing flows (NFs) for continuous $\boldsymbol{z}$, these approximations cannot perfectly represent all possible distributions. As a result, the lower bound may not be perfectly tight, introducing a gap between the estimated and true EIG (i.e., bias).
> >     For example, if the variational family is restricted to Gaussians, suboptimal interventions may yield approximately Gaussian posteriors, where the EIG lower bound is tight and produces a high value (e.g., 7). In contrast, optimal designs that generate more complex, expressive posteriors may result in poor Gaussian approximations, producing a lower EIG lower bound (e.g., 5) despite the true EIG being higher (e.g., 10).
> >     To address this bias, it is essential to use variational families that are more flexible and expressive. In our work, we employ NFs to better capture the posterior distribution.
> >
> > 10. __Advantage of $\pi$__:
> >     Using a policy network $\pi$ to design the next intervention based on historical samples is fundamental to making our framework both adaptive and non-myopic. Unlike traditional active learning methods, which compute each intervention design independently, $\pi$ learns to generate designs dynamically by considering historical data. Once trained, $\pi$ can efficiently produce intervention designs in real-time with just a forward pass, making it highly suitable for real-world deployment.
> >     Moreover, the policy network $\pi$ allows us to maximize the total Expected Information Gain (EIG) over a full sequence of $T$ interventions. Traditional active learning methods focus only on the next-step intervention, without accounting for the $T-1$ future steps, leading to a myopic approach. In contrast, our method optimizes the total EIG across all $T$ steps, ensuring a non-myopic strategy. We also theoretically demonstrate that the total EIG can be computed without explicitly evaluating intermediate posteriors, significantly improving the efficiency of training the policy network.
> >     By comparison, active learning methods, such as the CBED baselines used in our manuscript, require the explicit computation of intermediate posteriors at each step, which is computationally expensive. Our experimental results demonstrate that the proposed method not only provides non-myopic designs but also enables real-time decision-making for interventional experiments.
> >
> > Thank you again for your time and constructive feedback. We hope our clarifications address your concerns and provide a clearer understanding of our work.
> >
> > [1] Alex Greenfield, Aviv Madar, Harry Ostrer, and Richard Bonneau. Dream4: Combining genetic and dynamic information to identify biological networks and dynamical models. PloS one, 5(10):e13397, 2010.
> >
> > [2] Lars Lorch, Scott Sussex, Jonas Rothfuss, Andreas Krause, and Bernhard Schölkopf. Amortized inference for causal structure learning. Advances in Neural Information Processing Systems, 35:13104–13118, 2022.
> >
> > [3] Shen, Wanggang, Jiayuan Dong, and Xun Huan. "Variational sequential optimal experimental design using reinforcement learning." arXiv preprint arXiv:2306.10430 (2023).

---

> > > ### Comment · Reviewer_CYhw · 2024-11-25
> > >
> > > Thank you for the detailed response. While it partially addressed my concerns, I still find the lack of theoretical analysis on convergence to be a limitation. Additionally, there is room for improvement in the clarity of the writing. Considering the updates, I have increased my score to 5.

---

> > > > ### Author Response · Authors · 2024-12-03
> > > > **Official Comment by Authors**
> > > >
> > > > We sincerely appreciate your constructive feedback and the time you have dedicated. In response to your suggestion, we are conducting additional experiments with longer stages to empirically analyze the convergence results and will include these results in our next version.

---

### Official Review · Reviewer_6SUC · 2024-10-31

**Soundness:** 2
**Presentation:** 3
**Contribution:** 2
**Rating:** 6
**Confidence:** 4

**Summary:**

The authors develop a sequential experimental design framework to pick intervention targets and values to learn a particular causal quantity under a fixed experimental budget. Their method aims to provide an experimental design that aims to (1) maximize the information gain after the entire experimental budget instead of greedily selecting interventions at each step, (2) allow estimating different causal quantities directly without having to learn intermediate quantities (e.g., the graph), and (3) be amortized, that is, having a low computational cost in selecting the next intervention after each experiment.

To produce a policy, the authors use a transformer-based architecture that is trained using data-generating processes sampled from the prior and repeated “simulated” runs of experiment sequences. During deployment, the network does not need to be retrained after obtaining the data from each experiment. The training objective is a variational lower bound on the expected information gain over the complete experimental budget. The authors provide a proof on the validity of this bound.

The authors evaluate their method on a collection of synthetic data: synthetic causal models with different underlying graphs and structural assignments, and simulations of gene expression mechanisms in E-Coli and Yeast.

**Strengths:**

- The paper takes an innovative approach to providing an amortized experiment selection policy.
- The experiment section provides experiments to back up some of the claims of the paper, providing different data-generating models and quantities of interest. However, it lacks details (see weaknesses)
- Besides some issues with grammar (see Questions section), the text is easy to follow
- The figures (e.g., 1 and 2) are informative and nicely complement the text

**Weaknesses:**

In my opinion, the main weakness of the paper is that there are few details on the experimental setup, which is exacerbated by not providing code or artifacts to reproduce the results. There is also no reproducibility statement.

At best, this makes it difficult to perform an in-depth analysis of the results. At worst, it gives the impression that the authors picked some settings that are excessively favorable to their method and are trying to hide this fact. This overshadows the otherwise valuable contribution of the paper. In my opinion negative results are also valuable, and as long as the methodology is comprehensive this is not grounds for rejection.

I am more than happy to raise my score if these issues are addressed— also, if I missed this information, please let me know!

Let me turn this into some actionable feedback:

- For figures 3,4,8,9:  I apologize if I missed this, but are these results on a single data-generating process (i.e., ground-truth SCM) or an average over many graphs? If the results are over only one graph: how was this graph chosen? If you ran several examples and picked the one with the best results, you should say this and provide the results for a few other random examples in the appendix. Showing statistics (e.g. mean results + error bars) over a large number of randomly sampled data-generating processes would be the most informative and dispel any suspicions.
- For the in-silico experiments with (E. Coli / Yeast). You say ACED achieves almost perfect performance, but you give no details about the experimental setup, except that you use a 10-node graph. There is also no reference to the datasets you used.
  - How did you generate the data? Giving details would dispel the suspicion that you are cherry-picking. It would be good to provide some details such as the source of the gene regulatory network structure, the type of noise model used (if there is one), and any preprocessing steps applied to the data. You should also make the datasets publicly available.
  - What is the experiment horizon? Sample size of each experiment?
- How did you train Algorithm 1 in the experiments? In particular,
  - What is the prior used to train the network in Algorithm 1? Is it the exact same prior as the one used to generate the data-generating processes in Appendix 8?
  - What is the number of training samples n_env? I can imagine that if this is extremely large, you are able to reasonably cover the prior space. I don’t see this as a drawback, but it would be interesting to understand how the performance of the method is affected by this parameter, as it would be useful for a practitioner to know that they can improve performance by simpling elongating the training stage! As an idea, an ablation study showing how different values of n_env affect the performance, could provide valuable practical insights.
- Line 314: “We present results for ER graphs in the linear SCM setting”. However, in the results in Appendix 10, on non-linear SCMs for ER graphs, ACED performs close or worse than the random policy. This should be clearly stated in the main text, instead of hidden behind “the prior is more informative for SF graphs” in line 364. Admitting “mixed results” is in my opinion totally OK, and should not be a reason for rejection.

**Questions:**

- Throughout, the authors have a tendency to omit articles (a / the) in front of words. Here are some examples (I stopped after a few, so maybe check the whole text):
  - Line 053: “suboptimality in learning full causal model…” -> “suboptimality in learning A full causal model…” (or make models plural)
  - Line 302: “For causal reasoning task” -> “For THE causal reasoning task” (or make tasks plural)
  - Line 431: “Considering uniform intervention on node 1” -> “Considering uniform interventionS…” or “Considering A uniform intervention…”
  - Line 986: “over graph” -> “over the graph”?
  - Line 964: “Scale-free graph is characterized…” -> “Scale-free graphS ARE charachterized” or “A scale-free graph is…”
- Some weird sentences / typos:
  - Line 304: “[...] NMC could be efficient for estimating the shifted EIG than NFs” -> perhaps this should be “more efficient”?
  - Line 484: “[...] on Erdos-Renyi with 10 nodes” -> “on AN Erdos-Renyi graph with 10 nodes”
  - Line 485: “mechanism” -> “mechanismS”
- The sentence in Line 431 is unclear:
  - The enumeration [-5,4,...5] is a bit weird. Did you mean [-5,-4,...5]?
  - Also, what do you mean with values [-5,4,...5]. Is it that the intervention value is sampled uniformly at random from this range?
- Some minor points:
  - Line 282: Reference to (Erdos & Renyi, 1959) is strangely formatted -> perhaps something is off with the bibliography entry?
  - Line 311: You repeat Erdos-Renyi twice
  - Line 283: “various node sizes” -> I assume you mean number of nodes?
  - References to figures are not consistent, i.e., sometimes you write figure and sometimes Figure (e.g., figure 7 in line 485)
  - Subfigures in Figure 1 are labelled with double parenthesis, e.g., ((a)). In line 086 this also results in a double parenthesis when referencing it, i.e., “(Figure 1(b))”. An idea is to simply write (Figure 1b).

---

> ### Author Response · Authors · 2024-11-24
> **Response to Reviewer 6SUC (Part 1/2)**
>
> We thank the reviewer for the detailed and valuable feedback. We are committed to releasing the code publicly and will do so once it is fully prepared. We address each of your concerns and questions below:
>
> 1. __Experiments related Figures 3,4,8,9__:
> Due to space limitations, we have provided detailed statistics in tables in Section 10.4 of the Appendix for better illustration. For each case (e.g., 10-node linear SCMs with an Erdős–Rényi graph prior), we randomly sampled one SCM from the prior as the ground-truth SCM, ensuring it was not used during training. From this ground-truth SCM, we generated $n_{\text{obs}}$ observation samples to perform posterior inference (details in Section 8 of the Appendix), which served as the prior for training the policy network. The policy network was then trained using environments simulated from this prior. During evaluation, all methods interacted directly with the ground-truth SCM. The reported metrics are averaged over four random seeds used for initializing the policy and posterior networks. We acknowledge that incorporating multiple randomly sampled data-generating processes would provide a more robust evaluation and plan to explore this in future work.
>
> 2. __Settings of semi-synthetic GRN datasets__:
> In the semi-synthetic setting, we use the DREAM (Dialogue for Reverse Engineering Assessments and Methods) benchmarks [1], which are designed to evaluate computational methods for reverse engineering biological networks. These benchmarks provide realistic simulations of gene regulatory and protein signaling networks, generated using GeneNetWeaver v3.12. The simulator incorporates both ordinary differential equations (ODEs) and stochastic differential equations (SDEs) to model complex biological mechanisms and their inherent noise. For our experiments, we focus on two specific subnetworks from the DREAM benchmark: the E. coli and Yeast networks, each containing 10 nodes that represent the true causal graph. We simulate the mechanisms using the linear ANM setting with parameters $\theta \sim N(0, 2)$ and noise variances $\sigma\_i^2 \sim \text{InverseGamma}(4, 0.5)$.
>
> 3. __Algorithm training__:
> Due to space limits, we refer the details about the priors to the paragraph titled "Posterior inference with observational data" in Section 8 of the Appendix in the revised manuscript. We have updated the text to reference this when introducing the algorithm. Additionally, we refer to the details of the training in Section 10.1 of the Appendix in the revised manuscript. We agree that further ablation studies on $n\_{\text{envs}}$, $n\_{\text{step}}$, and $T$, particularly for nonlinear cases, could provide valuable insights, and we plan to include these in future work.
>
> 4. __Experimental results illustration__:
> We thank the reviewer for bringing up this important point. We have revised the manuscript to explicitly state the performance in nonlinear settings with both ER and SF graphs. Below, we provide additional explanations. In the context of Bayesian causal discovery, the goal of Bayesian Optimal Experimental Design is to design experiments that enable accurate estimation of the posterior distribution over causal graphs. In our study, we use AVICI [2] as the variational posterior model and simulate nonlinear structural causal models (SCMs) using a two-layer feed-forward neural network (FFN) with ReLU activation. Compared to linear cases, this setup increases the variability of simulated intervention samples from the prior, introducing higher variance in the EIG estimation and potentially requiring significantly more training steps for accurate estimation. We plan to further investigate this by conducting experiments with extended training steps for nonlinear cases in future work. Additionally, the prior $p(\boldsymbol{\theta} \mid G, \mathcal{D})$ used for simulating trajectories corresponds to the posterior conditioned on $\mathcal{D}$ and is trained using ELBO under a mean-field approximation. However, this approximation struggles to capture complex posterior dependencies, further contributing to inefficiencies in the simulation-based policy training process. To address this limitation, we propose exploring Stein Variational Gradient Descent (SVGD) as an alternative to ELBO to improve posterior quality and enhance performance in nonlinear settings.
>
> 5. __Enumeration $ [-5,-4,...5]$__:
> Thank you for pointing this out. To clarify, we compare the EIG estimates across all integer intervention values within the range $[-5, -4, ..., 5]$, resulting in a total of $11$ values. We have updated the problem setup in the revised manuscript to make this clearer.
>
> 6. __Typos__:
> We appreciate the reviewer pointing out the typos. We have corrected the mentioned errors in the updated manuscript and conducted a thorough review of the entire text.

---

> > ### Author Response · Authors · 2024-11-24
> > **Response to Reviewer 6SUC (Part 2/2)**
> >
> > Thank you again for your time and insightful feedback. We hope our clarifications address your concerns and provide a clearer understanding of our work.
> >
> >
> > [1] Alex Greenfield, Aviv Madar, Harry Ostrer, and Richard Bonneau. Dream4: Combining genetic and dynamic information to identify biological networks and dynamical models. PloS one, 5(10):e13397, 2010.
> >
> > [2] Lars Lorch, Scott Sussex, Jonas Rothfuss, Andreas Krause, and Bernhard Schölkopf. Amortized inference for causal structure learning. Advances in Neural Information Processing Systems, 35:13104–13118, 2022.

---

> ### Comment · Reviewer_6SUC · 2024-11-26
>
> I thank the authors for their clear answers and for including additional information about the experiments in the appendices. I also appreciate the revision of the text.
>
> While I appreciate that the authors include results for different random seeds of the method, I still have some reservations regarding the fact that the synthetic experiments are done on a single SCM. The fact that the authors don't share the code, artifacts, or provide a reproducibility statement is still problematic.
>
> However, in light of their changes, I am inclined to recommend acceptance and have increased my score.

---

> > ### Author Response · Authors · 2024-12-03
> > **Official Comment by Authors**
> >
> > We sincerely appreciate your valuable comments and recognition of our work. Thank you for your time and effort!

---

### Official Review · Reviewer_uT4Q · 2024-11-01

**Soundness:** 2
**Presentation:** 3
**Contribution:** 2
**Rating:** 5
**Confidence:** 3

**Summary:**

This paper introduces a novel approach called Adaptive Causal Experimental Design (ACED) for optimizing intervention selection in experimental design. The proposed framework generates non-myopic interventions, taking into account the effects on future experiments, unlike existing methods that optimize interventions in a greedy (myopic) manner via an offline policy network. ACED maximizes the Expected Information Gain (EIG) on causal quantities of interest, such as causal graph structure and specific causal effects, bypassing the need for intermediate posterior computations.

**Strengths:**

* Unlike previous methods, the proposed method uses the information from the history of experiments to select the best possible experiment for the next stage.
* The offline policy network significantly reduces the computational cost of the proposed framework during deployment.

**Weaknesses:**

* The training costs (e.g. in terms of compute and number of required interventional data points) of the offline policy model are not accounted for in the paper. While it is true that the proposed method requires significantly less computational budget during deployment, these training costs should not be completely overlooked.
* The method does not seem to scale well with a larger number of nodes e.g. it is evident in Figures 8 and 9 that DiffCBED starts to outperform the proposed method on both E-SHD and F1 score as the number of nodes increases.

**Questions:**

1. As mentioned in the weaknesses, Figures 8 and 9 show that other baselines start to outperform the proposed method by a considerable margin as n increases in both graph types, as well as in both linear and nonlinear cases. Could you please elaborate on this?
2. How many steps and samples at each step ($n_{step}$ and $T$) are used for training the offline policy model?
3. How large is the policy network e.g. in terms of number of parameters?
4. Can you explain how the proposed method performs under changes and distribution shifts compared to online methods? This is particularly important in dynamical systems where the underlying mechanisms or noise may change over time.
5. It seems like "D" is missing on the LHS of Equations 5 and 6.
6. How are the prior P(G|D) and the likelihood defined?
7. Given that the experiments are only conducted on synthetic data, can you elaborate on how the method can be used/extended to semi-synthetic/real-world settings (e.g. [1])?

[1] Greenfield, Alex, et al. "DREAM4: Combining genetic and dynamic information to identify biological networks and dynamical models." PloS one 5.10 (2010): e13397.

---

> ### Author Response · Authors · 2024-11-24
> **Response to Reviewer uT4Q (Part 1/2)**
>
> We thank the reviewer for the constructive and helpful feedback. We address each of your concerns and questions below:
>
> 1. __Training cost__:
> We present the training time of our proposed policy network across various synthetic datasets in the following table. For graphs with $d = 10$ and $d = 20$ nodes, we set the number of training steps $n_{\text{step}} = 10{,}000$, while for $d = 30$ nodes, we use $n_{\text{step}} = 15{,}000$. In each training step, $n_{\text{env}} = 10$ environments are sampled. Detailed training configurations are provided in Section 10.1 of the Appendix in the revised manuscript.
>
> | Prior                       | Training Time - 10 nodes | Training Time - 20 nodes | Training Time - 30 nodes |
> | --------------------------- | ------------------------ | ------------------------ | ------------------------ |
> | Erdos-Renyi + linear SCM    | 7.7h                     | 10.3h                    | 69.8h                    |
> | Scale-free + linear SCM     | 7.9h                     | 10.9h                    | 69.6h                    |
> | Erdos-Renyi + nonlinear SCM | 10.7h                    | 26.8h                    | 55.9h                    |
> | Scale-free + nonlinear SCM  | 10.6h                    | 25.0h                    | 51.5h                    |
>
> 2. __Worse performance for ACED for a larger number of nodes__:
>
> Thank you for pointing this out. First, the policy is designed to maximize the Expected Information Gain (EIG) across $T$ stages, with the expectation taken over all possible interventional samples. However, in the context of Bayesian causal discovery, as the number of nodes increases, the space of Directed Acyclic Graphs (DAGs) grows exponentially. This expansion necessitates a significantly larger number of training steps and training environments to fully recover the prior space, which is extremely challenging. To address these challenges, we plan to explore more efficient and generalizable architectures for scaling up and improving training efficiency in future work. Additionally, the prior $p(\boldsymbol{\theta} \mid G, \mathcal{D})$, which is used for simulating trajectories, is modeled as the posterior conditioned on $\mathcal{D}$. This prior is trained using the Evidence Lower Bound (ELBO) under a mean-field approximation in nonlinear cases. However, this approximation struggles to capture complex posterior dependencies, further contributing to inefficiencies in the simulation-based policy training process. To overcome these limitations, we propose investigating Stein Variational Gradient Descent (SVGD) as an alternative to ELBO in future work. SVGD has the potential to enhance posterior quality and improve the overall efficiency of the policy training process.
>
>
> 3. __Model hyperparameters__:
> We have added details about the hyperparameter configurations and specific model architecture in Section 10.1 of the Appendix in the revised manuscript.

---

> > ### Author Response · Authors · 2024-11-24
> > **Response to Reviewer uT4Q (Part 2/2)**
> >
> > 4. __Performance under distribution shifts__:
> > Thank you for highlighting this aspect. We have added a comparison between ACED and a random policy with a trained posterior network for causal discovery scenarios involving 10 nodes on Erdős–Rényi and scale-free graphs (details can be found in Section 10.2 of the Appendix in the revised manuscript). Here we consider the case where the noise distribution is changed from Gaussian to Gumbel, specifically,
> > $$
> >     X\_i = f\_i(\boldsymbol{X}\_{par(i)}, \boldsymbol{\theta}\_i) + \epsilon\_{X\_i}),
> >     \epsilon\_{X\_i} \sim N(0, \sigma\_i\^2)
> > $$
> > While the policies $\pi$ and the posterior networks $q_{\boldsymbol{\lambda}}(\cdot)$ for previous cases are trained with $\epsilon_{X_i} \sim N(0, 0.3^2)$, we evaluate their performance when applied to data generated with  $\epsilon_{X_i} \sim \text{Gumbel}(0, \sigma_i)$ and $\sigma_i^2 \sim \text{InverseGamma}(10, 1)$. To assess the robustness of the trained policies and posterior networks, we also train a separate $q_{\boldsymbol{\lambda}}(\cdot)$ using a random policy on data generated with the shifted noise distribution for causal discovery tasks involving 10 nodes on both Erdős-Rényi and Scale-Free graphs. The results are shown in the following table:
> >
> > | Number of stages | Metric | ACED (ER) | Random (ER) | ACED (SF) | Random (SF) |
> > |-----------------|--------|------------|-------------|------------|-------------|
> > | 2 | $\log q$ ($\uparrow$) | $-0.529 \pm 0.034$ | $\mathbf{-0.360 \pm 0.037}$ | $-0.457 \pm 0.017$ | $\mathbf{-0.337 \pm 0.055}$ |
> > | | $\mathbb{E}\text{-}\operatorname{SHD}$ ($\downarrow$) | $20.64 \pm 0.93$ | $\mathbf{14.99 \pm 1.31}$ | $17.05 \pm 0.51$ | $\mathbf{15.25 \pm 2.87}$ |
> > | | $F_1$ Score ($\uparrow$) | $0.477 \pm 0.072$ | $\mathbf{0.681 \pm 0.018}$ | $0.343 \pm 0.170$ | $\mathbf{0.549 \pm 0.128}$ |
> > | 4 | $\log q$ ($\uparrow$) | $-0.328 \pm 0.033$ | $\mathbf{-0.231 \pm 0.015}$ | $-0.296 \pm 0.022$ | $\mathbf{-0.189 \pm 0.047}$ |
> > | | $\mathbb{E}\text{-}\operatorname{SHD}$ ($\downarrow$) | $14.53 \pm 3.13$ | $\mathbf{11.03 \pm 1.41}$ | $11.65 \pm 1.78$ | $\mathbf{9.20 \pm 2.57}$ |
> > | | $F_1$ Score ($\uparrow$) | $0.662 \pm 0.077$ | $\mathbf{0.795 \pm 0.016}$ | $0.634 \pm 0.072$ | $\mathbf{0.764 \pm 0.078}$ |
> > | 6 | $\log q$ ($\uparrow$) | $-0.215 \pm 0.027$ | $\mathbf{-0.188 \pm 0.008}$ | $-0.181 \pm 0.027$ | $\mathbf{-0.133 \pm 0.028}$ |
> > | | $\mathbb{E}\text{-}\operatorname{SHD}$ ($\downarrow$) | $10.91 \pm 1.88$ | $\mathbf{8.62 \pm 1.33}$ | $7.87 \pm 3.57$ | $\mathbf{6.72 \pm 1.17}$ |
> > | | $F_1$ Score ($\uparrow$) | $0.778 \pm 0.023$ | $\mathbf{0.822 \pm 0.047}$ | $\mathbf{0.858 \pm 0.056}$ | $0.843 \pm 0.057$ |
> > | 8 | $\log q$ ($\uparrow$) | $\mathbf{-0.140 \pm 0.013}$ | $-0.167 \pm 0.007$ | $\mathbf{-0.097 \pm 0.022}$ | $-0.116 \pm 0.022$ |
> > | | $\mathbb{E}\text{-}\operatorname{SHD}$ ($\downarrow$) | $\mathbf{6.60 \pm 0.56}$ | $9.25 \pm 1.41$ | $\mathbf{5.13 \pm 2.46}$ | $6.32 \pm 1.08$ |
> > | | $F_1$ Score ($\uparrow$) | $\mathbf{0.885 \pm 0.042}$ | $0.829 \pm 0.029$ | $\mathbf{0.957 \pm 0.013}$ | $0.849 \pm 0.060$ |
> > | 10 | $\log q$ ($\uparrow$) | $\mathbf{-0.110 \pm 0.007}$ | $-0.157 \pm 0.007$ | $\mathbf{-0.066 \pm 0.013}$ | $-0.103 \pm 0.016$ |
> > | | $\mathbb{E}\text{-}\operatorname{SHD}$ ($\downarrow$) | $\mathbf{5.16 \pm 0.15}$ | $9.04 \pm 1.13$ | $\mathbf{2.52 \pm 0.678}$ | $6.23 \pm 0.49$ |
> > | | $F_1$ Score ($\uparrow$) | $\mathbf{0.913 \pm 0.030}$ | $0.835 \pm 0.040$ | $\mathbf{0.971 \pm 0.020}$ | $0.868 \pm 0.031$ |
> >
> > In future work, we plan to extend these results to evaluate the generalization performance of ACED against other online methods.
> >
> > 5. __Missing $\mathcal{D}$__:
> >     Thank you for pointing this out. We have revised the manuscript accordingly to ensure consistency.
> >
> > 6. __Prior and likelihood definition__:
> >     Thank you for pointing this out. We have added explanations for the prior $p(G \mid \mathcal{D})$ and $p(\boldsymbol{\theta} \mid \mathcal{D}, G)$ when introducing the algorithm, with further details provided in Section 8 of the Appendix. Additionally, we have revised Equation (2) in the manuscript to formally define the likelihood.
> >
> > Thank you again for your time and constructive feedback. We hope our clarifications address your concerns and provide a clearer understanding of our work.

---

> > > ### Comment · Reviewer_uT4Q · 2024-11-25
> > >
> > > Thank you for the clear responses. I appreciate the authors' efforts in providing additional experimental results on the model's performance under distribution shifts and the training times of the offline policy model. However, when considering both the offline training times of the policy network and the online deployment time of the proposed framework, I feel that the contribution of significantly lower computational cost during deployment becomes less compelling. I also believe there is still room for further improvement in the writing and flow of the paper. Considering the other reviews and the authors' responses, and the points mentioned above, I will maintain my current score.

---

> > > > ### Author Response · Authors · 2024-12-03
> > > > **Official Comment by Authors**
> > > >
> > > > We sincerely thank you for your constructive feedback, and we will continue to improve the writing and experiments based on your feedback.

---

### Official Review · Reviewer_JkFN · 2024-11-03

**Soundness:** 2
**Presentation:** 3
**Contribution:** 1
**Rating:** 3
**Confidence:** 3

**Summary:**

The paper presents ACED (Adaptive Causal Experimental Design), a method for sequential experimental design in causal discovery using reinforcement learning. The approach trains a transformer-based policy network to select interventions that maximize expected information gain for causal queries. The method aims to be non-myopic, computationally efficient at deployment, and flexible for different causal queries.

**Strengths:**

* Good technical writing and presentation.
* The technical execution is solid.
* Interesting extension to handle specific causal queries beyond structure learning.
* Novel mutation of the AVICI architecture to handle multimodality.

**Weaknesses:**

* Overlap with concurrent work (CAASL) that is not acknowledged or discussed, although cited. Please see next section for questions.

**Questions:**

The paper has significant overlap with CAASL (https://arxiv.org/abs/2405.16718v1) in several key aspects: Both use transformer-based policies with alternating attention and both train using RL (i understand this work is not branded as RL but it's a policy optimization method maximizing a reward based on AVICI model) to maximize information gain. Although ACED focuses also on causal queries beyond structure learning and uses a different variational approximations for the posterior, this work misses comparisson with CASSL (e.g. as a baseline) or a critical evaluation of the differences and the novelty. How does your work differ fundamentally from CAASL? Which components of your method are novel compared to CAASL?

Also, you compare with CBED line of work which is not sequential but batch experimental design, the main difference being that during batch selection there is no for feedback and thus can't adjust for the new information. I would argue that this might not be the most fair comparisson but i'm keen to hear your views on this.

In figure 4, the methods starting points have very large variance. I would expect for small number of steps (e.g. 2) to have relatively smaller differences. Do all methods use the same posterior model and hyperparameters during evaluation?

---

> ### Author Response · Authors · 2024-11-24
> **Response to Reviewer JkFN**
>
> We thank the reviewer for the constructive and helpful feedback. Below, we address your concerns and questions in detail:
>
> 1. __Overlap with CAASL__:
> Thank you for pointing this out; we would like to clarify. We did not include a comparison with CAASL in our work because the code was not publicly available at the time of submission. Although the authors have since kindly shared the code, additional effort is required to ensure a fair and meaningful comparison. Below, we highlight the key differences and outline our plans for future comparisons:
>
> * __Scope of the approach__: Our method provides a unified framework for designing optimal policies to reduce uncertainty for general causal queries, whereas CAASL focuses specifically on causal discovery.
>
> * __Reward computation__:
> CAASL computes rewards by sampling from the variational posterior $q$ and comparing the samples to the data-generating graph $G$. In contrast, our method evaluates the density of $G$, which is potentially more memory- and computationally-efficient.
>
> * __Training approach__:
> CAASL uses Soft Actor-Critic (SAC) to train the policy, which increases training time but does not require a differentiable reward function. Our approach, on the other hand, relies on a differentiable likelihood for policy optimization.
>
>     Additionally, there are other setup differences, such as the inclusion of initial observations $\mathcal{D}$, the data-generating process, the posterior inference algorithm, and whether the pretrained AVICI model is fine-tuned. In future work, we will ensure a fair and comprehensive comparison with CAASL by carefully aligning these experimental setups.
>
>
> 2. __Batch experimental design for CBED__:
>     In our implementation of the CBED approaches [1, 2], we perform single-target interventions iteratively, followed by posterior inference. Specifically, while our policy is trained for $T$ stages with one intervention target per stage, CBED adopts a greedy design strategy. It executes one intervention target at each stage and uses intermediate feedback to inform the subsequent interventions, rather than conducting $T$ interventions in a single batch.
>
> 3. __Large variance in the starting point \& posterior for each method__:
>     The large variance arises partly because other approaches perform Bayesian Optimal Experimental Design (BOED) in an active, greedy manner, optimizing for immediate information gains at early stages. In contrast, the ACED policy is designed to maximize the Expected Information Gain (EIG) across all $T$ stages, following a non-myopic approach that may sacrifice early-stage rewards for greater long-term benefits.
>     Additionally, the posterior reward model in our method uses max-pooling to handle inputs with varying numbers of intervention samples but is trained on intervention samples with a fixed shape of $n_{\text{int}} \times T$. This training setup can sometimes lead to suboptimal performance at early stages (small $T$).
>     Finally, unlike other methods that rely on DiBS [3] for posterior inference, our approach simultaneously trains the policy and posterior model. This distinction has been noted in the Metrics paragraph of Section 5.
>
> Thank you again for your time and helpful feedback. We hope our clarifications address your concerns and provide a clearer understanding of our work.
>
> [1] Tigas, Panagiotis, Yashas Annadani, Andrew Jesson, Bernhard Schölkopf, Yarin Gal, and Stefan Bauer. "Interventions, where and how? experimental design for causal models at scale." Advances in neural information processing systems 35 (2022): 24130-24143.
>
> [2] Tigas, Panagiotis, Yashas Annadani, Desi R. Ivanova, Andrew Jesson, Yarin Gal, Adam Foster, and Stefan Bauer. "Differentiable multi-target causal Bayesian experimental design." In International Conference on Machine Learning, pp. 34263-34279. PMLR, 2023.
>
> [3] Lorch, Lars, Jonas Rothfuss, Bernhard Schölkopf, and Andreas Krause. "Dibs: Differentiable Bayesian structure learning." Advances in Neural Information Processing Systems 34 (2021): 24111-24123.

---

> > ### Comment · Reviewer_JkFN · 2024-11-26
> >
> > ## tl;dr
> > I maintain my current score. While this work makes some contributions, I find significant overlap with CAASL that needs to be addressed. The core components - the Barber-Agakov reward and transformer architecture (based on Kossen, 2021), the use of AVICI model - are shared between the works. The main differences lie in implementation details: ACED uses a differentiable reward formulation whereas CAASL uses a non-differentiable approach, and ACED jointly trains the policy and posterior while CAASL uses a pretrained AVICI model. These differences, while interesting, represent smaller advances than currently claimed.
> >
> > ## Full response
> > While the paper positions itself beyond causal discovery, a substantial portion of the experimental evaluation (including SHD metrics and choice of baselines) addresses causal discovery tasks. To provide a complete picture, CAASL should be included as a baseline and the relationship between the works should be explicitly discussed. The similarities in foundational components (reward structure, architecture) should be acknowledged, along with a clear articulation of the technical differences - namely the differentiable reward formulation and joint training approach in ACED versus CAASL's non-differentiable implementation with a pretrained model. The relative advantages and tradeoffs between these approaches deserve discussion.
> > The paper would benefit from expanding on aspects truly unique to ACED, such as the benefits enabled by the differentiable formulation and joint training, as well as applications to causal queries beyond discovery. However, these incremental advances, while valuable, do not warrant raising the current score or changing the recommendation.
> >
> > ### minor points
> > References 2024a and 2024b appear to be duplicate entries.

---

> > > ### Author Response · Authors · 2024-12-03
> > > **Official Comment by Authors**
> > >
> > > Thank you for your additional comments. We would like to address them as follows:
> > >
> > > 1. __Regarding the overlap and comparison with CAASL__:
> > > While ACED and CAASL share similarities in the context of causal discovery, we emphasize that ACED's formulation offers greater flexibility, addressing not only causal discovery but also broader causal queries. This flexibility represents a significant advancement, as highlighted by recent works focusing on QoI-oriented Bayesian Optimal Experimental Design (BOED) [1] and active BOED for causal queries [2]. Regarding the inclusion of CAASL as a baseline, we note that at the time of submission, CAASL was available only as an arXiv preprint without publicly accessible code. Although the authors kindly shared their code during the rebuttal period, ensuring a fair comparison required substantial adjustments due to differences in setup. We plan to include the results of CAASL in our comparison in the next version of the manuscript.
> > >
> > > 2. __Typos__: Thank you for pointing this out. We have corrected them in the revised manuscript.
> > >
> > > Thank you again for your time and engagement!
> > >
> > > [1] Smith, Freddie Bickford, Andreas Kirsch, Sebastian Farquhar, Yarin Gal, Adam Foster, and Tom Rainforth. "Prediction-oriented Bayesian active learning." In International Conference on Artificial Intelligence and Statistics, pp. 7331-7348. PMLR, 2023.
> > >
> > > [2] Toth, Christian, Lars Lorch, Christian Knoll, Andreas Krause, Franz Pernkopf, Robert Peharz, and Julius Von Kügelgen. "Active Bayesian causal inference." Advances in Neural Information Processing Systems 35 (2022): 16261-16275.

---

### Official Review · Reviewer_m3o9 · 2024-11-05

**Soundness:** 3
**Presentation:** 2
**Contribution:** 4
**Rating:** 6
**Confidence:** 2

**Summary:**

Adaptive Causal Experimental Design (ACED) is an algorithm that chooses a sequence of experiments designed to maximize expected information gain (EIG) for a variety of different possible causal questions. In contrast to current algorithms, it is non-myopic (maximizes the EIG for a sequence of experiments rather than for the next experiment), is adaptive in real-time, and works on a variety of different causal questions, including constructing causal graphs and calculating the effects of interventions. It uses variational methods to lower bound the EIG for a given policy. Experiments indicates that the runtime of ACED is much faster than various other state of the art experimental design algorithms, and is more accurate in SHD and F1 score on reconstructing graphs in linear Gaussian models.

**Strengths:**

The strengths of the article are that it describes an algorithm that has major advantages over existing algorithms.
1. It is non-myopic.
2. It is real-time adaptive.
3. It is flexible which causal questions it can answer. l
4.It performs significantly better on SHD and F1 score in graph reconstruction on linear Gaussian data as the number of stages increases on simulated data.
5. It  performs significantly better on SHD and F1 score in graph reconstruction on two read Genetic Regulatory Networks.
6. It is much faster than alternative algorithms such as DiffCBED and SoftCBED.

**Weaknesses:**

1. A major advantage the authors claim for ACED is its speed. But they don't put their discussion of speed into the main body of the paper, even though there is room for it. In Figure 10, the X axis is deployment time, with no indication of what the units are. The Y axis is labelled variable dimensions. I am not sure what variable dimension refers to - number of variables? The ACED line has 0 variable dimensions for every Deployment time, while the diffBCED and Soft-BCED lines have variable dimension that start in the thousands and increase over Deployment time. The accompanying text say that this shows that ACED has significantly faster inference times, but it is hard to see exactly what the graph indicates or how it shows that ACED is faster. I am unclear about how many variables ACED can be applied to, and how long that would take
2. The differences between random choice of experiment and ACED are quite small in many of the non-linear cases, and in some cases even after a large number stages, the random experiment does better.
3. Points 1 and 2 raise the question of whether one would be better off using random experiments for large numbers of variables in non-linear cases.
4. The authors say that they took are to take care that the structural causal models that they generate are not identifiable from observational data. However, in the linear case, the data was generated with random noises all of which had equal variance. This is known to provide extra information and make the graph identifiable without doing any experiments. In the non-linear case they generated additive noise models, which are also known to be identifiable except in special circumstances (e.g. linear Gaussian.) See Zhang and Hyvaarinen in "On the Identifiability of the Post-Nonlinear Causal Model".
5. There are a number of minor issues that make the paper difficult to read:
   a. On line 187 p(M|D) is introduced with no explanation of what M is.
   b. Results are reported for two "realistic" data sets from yeast and E. coli. There is no description of these
       data sets at all - how many variables, how dense the graphs were, what functions related parents to
       children, etc., or in what sense they are "realistic".
   c. The paper does not define RELU, FFN.

**Questions:**

1. Where is the density of the simulated graphs specified? Without that, it is hard to tell how hard the problem is.
2. The authors make the following statement: "Although some baselines achieve a low E-SHD as the number of nodes increases, this is likely due to the posterior inference model DiBS converging to low-entropy solutions, which tend to predict only a few edges. When considering both E-SHD and F1-score, ACED outperforms the baselines significantly". E-SHD and F1-score are measuring very similar things and ought to be very highly correlated. Please explain how taking both into account would lead to a different conclusion than only taking E-SHD into account.
3. There are a number of different related ways of defining structural Hamming distance (for example how to treat a reversed edge.) Please specify more exactly how it was calculated.
4. In equations 3 and 4, h_t = {xi_{1:t}, x_{1:t}}. So I thought that would mean h_{t-1} =  {xi_{1:t-1}, x_{1:t-1}}. But then h_{1:t-1} also occurs in the equations. If h_{t-1} already contains all the information about xi and x from 1 to t-1, what is h_{1:t-1} add to h_{t-1}?
5. Could you expand the dicussion of the setup in 4.2?  What are n and d in Figure 2? Why is L = 4? A little more detail about what each part is doing would be helpful.
6. For the causal reasoning task, in the linear Figure 7 why not compare the estimated effects of the interventions directly, instead of log q?
7. How were the number of stages chosen? Is that limited by how long each stage took? For the larger graphs it looks like doing more stages would be helpful.

---

> ### Author Response · Authors · 2024-11-24
> **Response to Reviewer m3o9 (Part 1/2)**
>
> We sincerely thank the reviewer for their kind assessment and insightful feedback. We address each of your concerns and questions below:
>
> 1. __The deployment time comparison__: Thank you for your suggestion. We have updated Figure 8 (originally Figure 10) to a bar plot for improved clarity. The figure now illustrates the deployment times of our proposed method, ACED, compared to two baselines, DiffCBED and SoftCBED, across three variable dimensions (10, 20, and 30) on linear synthetic SCMs with Erdős–Rényi graphs. The deployment times are plotted on a logarithmic scale to capture the substantial differences in magnitudes between the methods. As shown in the bar plot, ACED achieves significantly faster deployment times than the baseline methods.
>
> 2. __Comparison to the random baseline in the nonlinear cases__: The reviewer raises an excellent point. In the context of Bayesian causal discovery, the goal of Bayesian Optimal Experimental Design is to design experiments that enable accurate estimation of the posterior distribution over causal graphs. In our study, we use AVICI [1] as the variational posterior model and simulate nonlinear structural causal models (SCMs) using a two-layer feed-forward neural network (FFN) with ReLU activation. This setup can increase the variability of simulated intervention samples from the prior compared to linear cases, introducing higher variance in the EIG estimation and potentially requiring significantly more training steps for accurate estimation. We plan to investigate this further by conducting experiments with extended training steps for nonlinear cases in future work. Additionally, the prior $p(\boldsymbol{\theta} \mid G, \mathcal{D})$ used for simulating trajectories corresponds to the posterior conditioned on $\mathcal{D}$ and is trained using ELBO under a mean-field approximation in nonlinear cases. This approximation struggles to capture complex posterior dependencies, further contributing to inefficiencies in the simulation-based policy training process. To address this limitation, we propose exploring Stein Variational Gradient Descent (SVGD) as an alternative to ELBO to improve posterior quality in future work.
>
> 3. __Identifiability__:
> Thank you for raising the concern about identifiability in the linear SCM setting. You are correct that a linear additive noise model (ANM) with equal Gaussian noise is identifiable from observational data. However, in the nonlinear SCM setting, we use a two-layer neural network with an FFN as the data-generating mechanism between nodes and their parents. Since neural networks are generally non-invertible, the nonlinear case is non-identifiable from observational data alone, as shown in [2]. To address this, we will include additional experiments in the linear setting, such as the linear ANM with non-equal Gaussian noise, which is non-identifiable from observational data. We also revised the manuscript to clarify that "carefully designed interventions facilitate the identifiability of the causal graph."
>
> 4. __$\mathcal{M}$ in Equation 9 (line 187)__:
> Thank you for pointing this out. $\mathcal{M}$ is defined as $\mathcal{M} = {G, \boldsymbol{\theta}}$, representing the causal graph $G$ and its associated mechanisms parameterized by $\boldsymbol{\theta}$. We have explicitly added this definition in the revised manuscript for clarity.
>
> 5. __Settings of semi-synthetic GRN datasets__:
> In the semi-synthetic setting, we use the DREAM (Dialogue for Reverse Engineering Assessments and Methods) benchmarks [3], which are designed to evaluate computational methods for reverse engineering biological networks. These benchmarks provide realistic simulations of gene regulatory and protein signaling networks, generated using GeneNetWeaver v3.12. The simulator incorporates both ordinary differential equations (ODEs) and stochastic differential equations (SDEs) to model complex biological mechanisms and their inherent noise. For our experiments, we focus on two specific subnetworks from the DREAM benchmark: the E. coli and Yeast networks, each containing 10 nodes that represent the true causal graph. We simulate the mechanisms using the linear ANM setting with parameters $\theta \sim N(0, 2)$ and noise variances $\sigma_i^2 \sim \text{InverseGamma}(4, 0.5)$.
>
> 6. __Density of the simulated graph__:
> For graphs with $d = 10$ nodes, we simulate graphs with an expected $2d$ edges, while for graphs with $d = 20$ or $d = 30$ nodes, we simulate graphs with an expected $d$ edges.

---

> ### Author Response · Authors · 2024-11-24
> **Response to Reviewer m3o9 (Part 2/2)**
>
> 7. __$\mathbb{E}$-SHD and $F_1$-score__:
> To clarify, we use DiBS [4] as the posterior inference model for the DiffCBED and SoftCBED baselines, consistent with prior work \cite{tigas2022interventions}. In low-data regimes, DiBS tends to produce low-entropy solutions, often sampling graphs with few or no edges. While this behavior aligns with the sparsity of the true graph and results in a low $\mathbb{E}$-SHD, it also leads to a low expected $F_1$ score because most edges are predicted as non-existent. Intuitively, an empty graph can achieve a relatively low $\mathbb{E}$-SHD by avoiding incorrect edges, but it will result in an expected $F_1$-score of 0, as no true edges are predicted across posterior samples.
>
> 8. __SHD definition__:
> In our implementation, we calculate the Structural Hamming Distance (SHD) using adjacency matrices, treating reversed edges as a distance of $1$.
>
> 9. __$h\_{t-1}$ definition__:
> Thank you for pointing this out. Yes, $h_{t-1}$ is equivalent to $h_{1:t-1}$, and we have updated the notations in the revised manuscript for consistency.
>
>
> 10. __Expand the setup in Section 4.2__:
> We have revised Section 4.2 to include additional details and provided further information about the policy network architecture in Appendix 10.1 of the revised manuscript. In this setup, $n$ represents the total number of interventions performed over an entire trajectory (i.e., $n_{\text{int}} \times T$), and $d$ is the number of variables in the system (or nodes in the causal graph). The parameter $L = 4$ is predetermined; while a larger $L$ could increase expressive power, it would also result in higher computational costs. We refer to $L$ as a pre-defined hyperparameter for our policy network.
>
> 11. __Why not compare the estimated effects of the interventions directly but using $\log q$__:
> The estimated effects of intervention is a distribution $p(\boldsymbol{z}|\mathcal{D}, \boldsymbol{h}\_{T})$ that entails the posterior predictive integration over $G$ and $\boldsymbol{\theta}$ as shown in equation (6). However, the true posterior $p(G|\mathcal{D}, \boldsymbol{h}\_{T})$ is intractable when the number of nodes is greater than 6 in general.
> In addition, the problem statement is to find the policy $\pi$ that maximizes EIG on $\boldsymbol{z}$ given $\mathcal{D}$ and $ \boldsymbol{h}\_{T}$ as in equation (8), which is hard to estimate in general while its lower bound given in equation (9) is tightened when $    \mathbb{E}\_{p(\boldsymbol{\mathcal{M}}|\mathcal{D})p(\boldsymbol{h}\_T|\mathcal{D}, \boldsymbol{\mathcal{M}}, \pi) p(\boldsymbol{z}|\mathcal{D}, \boldsymbol{\mathcal{M}}) } \log q\_{\boldsymbol{\lambda}}(\boldsymbol{z} |\mathcal{D},  f\_{\boldsymbol{\phi}}(\boldsymbol{h}\_T))$ is maximized. We have added explanations on why we are using $\log q$ when introducing the metrics in the revised manuscript.
>
>
> 12. __Number of stages__:
> Thank you for raising this important point. Currently, the number of stages is predetermined before the experiments and is influenced by factors such as the experimental budget, training time, and the number of nodes in the graph (since larger graphs generally require more interventions for causal discovery). For future work, we plan to improve the robustness of the policy across stages, enabling it to adapt to scenarios where immediate budget limitations prevent additional experiments during deployment.
>
> Thank you again for your time and valuable feedback. We hope our clarifications address your concerns and provide a clearer understanding of our work.
>
>
> [1] Lars Lorch, Scott Sussex, Jonas Rothfuss, Andreas Krause, and Bernhard Schölkopf. Amortized inference for causal structure learning. Advances in Neural Information Processing Systems, 35:13104–13118, 2022.
>
> [2] Kun Zhang and Aapo Hyvarinen. On the identifiability of the post-nonlinear causal model.
> arXiv preprint arXiv:1205.2599, 2012.
>
> [3] Alex Greenfield, Aviv Madar, Harry Ostrer, and Richard Bonneau. Dream4: Combining genetic and dynamic information to identify biological networks and dynamical models. PloS one, 5(10):e13397, 2010.
>
> [4] Lorch, Lars, Jonas Rothfuss, Bernhard Schölkopf, and Andreas Krause. "Dibs: Differentiable bayesian structure learning." Advances in Neural Information Processing Systems 34 (2021): 24111-24123.

---

> > ### Comment · Reviewer_m3o9 · 2024-11-25
> > **Clarifications**
> >
> > Thanks you for the clarifications and answering my questions. I will consider my rating in light of your answers, and the rest of the discussion with other reviewers.

---

> > > ### Author Response · Authors · 2024-12-03
> > > **Official Comment by Authors**
> > >
> > > We sincerely appreciate your insightful comments and acknowledgment of our work. Thank you for your time!

---

### Meta-Review · Area_Chair_tvu5 · 2024-12-21

**Metareview:**

As pointed out by one of the reviewers, there is a very related paper with similar methodology that the authors did not include. Based on their rebuttal, it seems that this was a deliberate choice by the authors as they say they chose to not compare because code was not available. Given the circumstances I explain below this is not a valid reason. This paper was available in May and the authors had time to incorporate it into their submission due October. We understand that the code was recently shared by the authors, which means the other set of authors would have shared the code had the authors reached out earlier.

It could very well have been an independent discovery of the same idea but since authors seem to have been aware of this paper (they cited it), they should have at least discussed these substantial similarities, as pointed out by one reviewer. Authors should incorporate the comments by all reviewers (some others are also quite valuable around clarity, emphasizing the right performance metrics in the main paper and so on) and clearly establish the differences with this method.

I have checked their rebuttal and the authors suggest that the proposed method provides greater flexibility in answering questions other than causal discovery. This is a great direction and would clearly establish the delta. But it requires a significant rewrite and repositioning of the paper.

**Additional Comments On Reviewer Discussion:**

The critical reviewer is convinced that the existing work carries significant similarities and should be compared against. Other reviewers, even those in favor of acceptance had important concerns, use of a single SCM in experiments, and the improvement in computation cost are questioned.

---

### Decision · Program_Chairs · 2025-01-22

Reject